# Pan-cancer analysis reveals TAp63-regulated oncogenic lncRNAs that promote cancer progression through AKT activation

Marco Napoli [1,2], Xiaobo Li[1,2], Hayley D. Ackerman[1,2], Avani A. Deshpande[1,2], Ivan Barannikov[1,2], Marlese A. Pisegna[1,2], Isabelle Bedrosian[3], Jürgen Mitsch[4,5], Philip Quinlan [4,5], Alastair Thompson[6], Kimal Rajapakshe[7], Cristian Coarfa [7], Preethi H. Gunaratne[8], Douglas C. Marchion[9], Anthony M. Magliocco[9], Kenneth Y. Tsai [2,9,10] & Elsa R. Flores [1,2✉]

The most frequent genetic alterations across multiple human cancers are mutations in *TP53* and the activation of the PI3K/AKT pathway, two events crucial for cancer progression. Mutations in *TP53* lead to the inhibition of the tumour and metastasis suppressor *TAp63*, a p53 family member. By performing a mouse-human cross species analysis between the *TAp63* metastatic mammary adenocarcinoma mouse model and models of human breast cancer progression, we identified two TAp63-regulated oncogenic lncRNAs, *TROLL-2* and *TROLL-3*. Further, using a pan-cancer analysis of human cancers and multiple mouse models of tumour progression, we revealed that these two lncRNAs induce the activation of AKT to promote cancer progression by regulating the nuclear to cytoplasmic translocation of their effector, WDR26, via the shuttling protein NOLC1. Our data provide preclinical rationale for the implementation of these lncRNAs and WDR26 as therapeutic targets for the treatment of human tumours dependent upon mutant *TP53* and/or the PI3K/AKT pathway.

[1] Department of Molecular Oncology, H. Lee Moffitt Cancer Center and Research Institute, Tampa, FL 33612, USA. [2] Cancer Biology and Evolution Program, H. Lee Moffitt Cancer Center and Research Institute, Tampa, FL 33612, USA. [3] Department of Surgical Oncology, The University of Texas M.D. Anderson Cancer Center, Houston, TX 77030, USA. [4] Advanced Data Analysis Centre, Nottingham NG7 2RD, UK. [5] School of Computer Sciences University of Nottingham, Nottingham NG7 2RD, UK. [6] Department of Surgery, Baylor College of Medicine, Houston, TX 77030, USA. [7] Department of Molecular and Cellular Biology, Baylor College of Medicine, Houston, TX 77030, USA. [8] Department of Biology and Biochemistry, University of Houston, Houston, TX 77004, USA. [9] Department of Anatomic Pathology, H. Lee Moffitt Cancer Center and Research Institute, Tampa, FL 33612, USA. [10] Department of Tumour Biology, H. Lee Moffitt Cancer Center and Research Institute, Tampa, FL 33612, USA. ✉email: elsa.flores@moffitt.org

Cancer metastasis is the leading cause of death in cancer patients[1]. Multiple pathways have been found to increase cancer progression and metastasis including the activation of the PI3K/AKT pathway[2] and the gain-of-function mutation of the tumour suppressor TP53[3], which are the two most frequent driving mutations in a broad variety of human cancers[4]. Therefore, investigating the mechanistic interplay between these pathways is of the utmost importance for the identification of novel therapeutic opportunities against the progression of metastatic cancers.

One of the mechanisms by which mutant TP53 exerts its gain of function is through the inhibition of the p53 family member and p63 isoform, TAp63[3]. We previously reported that TAp63 is a crucial tumour and metastasis suppressor. Mice lacking TAp63 (TAp63$^{-/-}$) develop highly metastatic tumours, with a large proportion being mammary adenocarcinomas that metastasize to the lung, liver, and brain[5]. Moreover, deletion of TAp63 in murine and human mammary epithelial cells (MECs) triggers their transformation into tumour initiating cells[6], which give rise to mammary adenocarcinomas metastasizing to distant sites[5]. The essential role of the tumour suppressive activity of TAp63 in human breast cancers is evident due to the inverse correlation of its expression with tumour grade[5].

The tumour and metastatic suppressive activity of TAp63 relies on the transcriptional regulation of gene expression and, until now, TAp63 has been shown to control the expression of protein-coding genes, including Dicer, and miRNAs[5–10]. Here, we demonstrate that TAp63 also governs the expression of long non-coding RNAs (lncRNAs), and notably that the levels and functional activities of two of these TAp63-regulated oncogenic lncRNAs or TROLLs correlate with the progression and tumour grade of a wide variety of human cancers. Using breast cancer as a model system and then extending our findings using a pan-cancer approach including xenograft mouse models, TCGA datasets, and 723 clinical cases, we provide molecular and functional evidence that the tumorigenic and metastatic potential of these lncRNAs is mediated by one of their interacting proteins, WDR26. The cytoplasmic localization of WDR26, which we found to be typical of advanced cancers, is controlled by TROLL-2 and TROLL-3 via the shuttling protein NOLC1 and is required for the pro-oncogenic and metastatic activities of WDR26, including the interaction with AKT and the induction of its activating phosphorylation on Ser473. The physical and functional interaction between the two TROLLs and WDR26 is particularly significant for basal-like breast cancers and melanomas, where high levels of these lncRNAs as well as high levels of TROLL-3 and WDR26 correlate with poor prognosis. Taken together, our findings identify a crucial mechanism for the activation of the AKT pathway through TAp63-regulated lncRNAs (TROLLs) and pave the way for more effective therapies against metastatic cancers with alterations in TP53 and hyperactivation of the PI3K/AKT pathway.

## Results

**TAp63 regulates lncRNAs in breast cancer progression.** TP53 missense mutations are the most frequent genetic alterations in breast cancers[11] and inactivate the tumour and metastasis suppressor TAp63[3]. Importantly, we have previously shown that loss of TAp63 leads to the onset of highly metastatic mammary adenocarcinomas to distant sites, making this a faithful mouse model of human metastatic breast cancers[5]. By performing RNA-seq (RNA-sequencing) analysis of wild-type (WT) and TAp63$^{-/-}$ MECs[6], we found that TAp63$^{-/-}$ MECs contained 591 lncRNAs that were differentially expressed compared to WT MECs. To determine whether these mouse lncRNAs had human orthologs

involved in breast cancer formation and progression, we used a locus conservation approach[12] to compare the differentially expressed mouse lncRNAs to human lncRNAs differentially expressed in the MCF10A breast cancer progression model[13,14], comprised of four cell lines: (i) MCF10A (normal mammary epithelial cells); (ii) AT1 (transformed); (iii) DCIS (tumorigenic); and (iv) CA1D (metastatic cells). This strategy allowed us to identify 9 TAp63-regulated oncogenic lncRNAs or TROLLs in mouse and human breast cancer progression (Fig. 1a, b). Importantly, one of the identified TROLLs was MALAT1, previously demonstrated to promote different metastatic tumour types in humans[15–17], including breast cancer where high levels of this lncRNA correlate with higher risk of relapse and reduced overall survival[18–20]. We performed qRT-PCR and found that 6 TROLLs were upregulated and three were downregulated in TAp63$^{-/-}$ MECs compared with WT MECs (Fig. 1c), and similarly their human orthologs were differentially expressed in the MCF10A breast cancer progression model cell lines (Supplementary Fig. 1a). To assess whether expression of these 9 human orthologs is TAp63-dependent, we targeted TAp63 in the metastatic mammary adenocarcinoma cell line, CA1D, via a doxycycline-inducible CRISPR/Cas9 system[21]. After 6 days of induction, we achieved a 45% cleavage efficiency in the TAp63 locus (Supplementary Fig. 1b) with a concurrent downregulation of TAp63 mRNA levels (Supplementary Fig. 1c). Reduced levels of TAp63 were associated with the upregulation of 6 lncRNAs and the downregulation of three lncRNAs (Fig. 1d), similar to what we observed for their murine orthologs in TAp63$^{-/-}$ vs. WT MECs. Conversely, the overexpression of TAp63 in both WT MECs and in CA1D cells was associated to the opposite change in the expression of these nine TROLLs (Supplementary Fig. 1d–g). To verify whether the nine human lncRNAs are bona fide direct target genes of TAp63, we analysed their respective promoters for possible p63 response elements and found putative TAp63 binding sites in all of these promoters (Supplementary Fig. 1h). Chromatin immunoprecipitation (ChIP) experiments revealed TAp63 recruitment to these sites (Supplementary Fig. 1i). These results indicate that TAp63 directly binds these promoters to transcriptionally regulate the expression of these nine conserved lncRNAs.

We previously demonstrated that TAp63 suppresses metastatic cancer by regulating a transcriptome that halts cancer migration and invasion[5,6]. To determine whether these nine conserved lncRNAs are involved in these biological properties associated with metastasis, CA1D cells were transfected with siRNAs targeting these lncRNAs, and cell migration and invasion were assessed. Down-regulation of these lncRNAs severely affected the migratory and invasive potential of CA1D cells (Fig. 1e, f) while having little effect on cell proliferation and survival (Fig. 1g, h). Indeed, only the silencing of one lncRNA (TROLL-2) decreased cell proliferation assessed by 5-ethynyl-2′-deoxyuridine (EdU) incorporation (Fig. 1g), and the silencing of 3 of them (TROLL-2, TROLL-5 and TROLL-8) caused a slight yet significant increase in the percentage of apoptotic cells, as quantified by annexin V staining (Fig. 1h). These findings indicate that TAp63 directly regulates the expression of conserved lncRNAs, which in turn are required for the efficient migration and invasion of mammary adenocarcinomas.

**TROLL-2 and TROLL-3 are markers of breast cancer progression.** To assess the relevance of the identified lncRNAs in human breast cancers, we decided to focus our attention on two lncRNAs, TROLL-2 and TROLL-3, since they are the only two among the nine TROLLs to be divergent, i.e. lncRNAs transcribed on the opposite strand compared to a nearby protein-coding gene

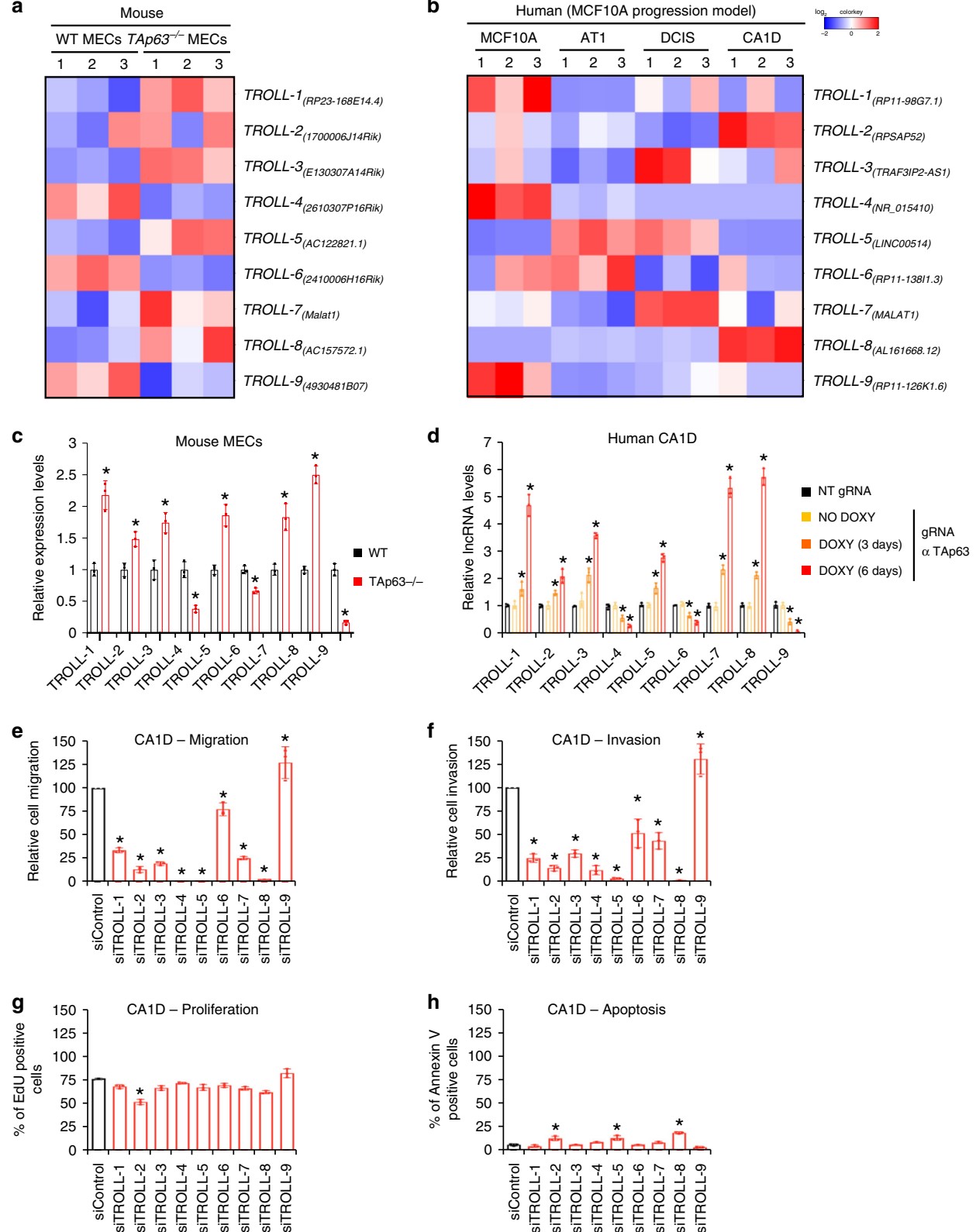

**Fig. 1 TAp63 regulates lncRNAs in breast cancer progression. a** Heatmap of the nine conserved lncRNAs differentially expressed in WT and *TAp63⁻/⁻* mammary epithelial cells (MECs). **b** Heatmap of the nine conserved lncRNAs differentially expressed in the MCF10A human breast cancer progression model. **c** qRT-PCR of the nine conserved lncRNAs in WT and *TAp63⁻/⁻* MECs. **d** qRT-PCR of the nine conserved lncRNAs in CA1D cells infected with either non targeting (NT) or *TAp63* targeting gRNA. 3 days and 6 days indicate the time of Cas9 induction via doxycycline. **e–h** Quantification for cell migration (**e**), cell invasion (**f**), EdU incorporation (**g**), and annexin V positivity (**h**) of CA1D cells transfected with siRNAs against the indicated lncRNAs. Graphs represent the individual data points, mean ± SD of three independent experiments. *P* value (*$p < 0.05$) was calculated by two-tailed unpaired Student's *t*-test. Source data are provided as Source data file.

and generally sharing similar functions in line with the guilt-by-association principle[22,23]. Intriguingly, their respective antisense protein coding genes (HMGA2 for *TROLL-2* and TRAF3IP2, also known as ACT1, for *TROLL-3*) are both known oncogenes supporting tumour growth and dissemination[24,25]. To assess whether these two lncRNAs were expressed in human invasive breast cancers, we performed in situ hybridization (ISH) for *TROLL-2* and *TROLL-3* in a breast cancer tissue microarray (TMA) with 45 samples including normal breast tissue, lobular hyperplasia, DCIS, and invasive breast cancer biopsies. Notably, we found that the levels of both lncRNAs were undetectable in normal breast tissue and increased with breast cancer progression with the highest levels observed in invasive breast cancer samples (Fig. 2a, b). Interestingly, the expression of *TROLL-2* and *TROLL-3* positively correlated in these samples (Supplementary Fig. 2a), indicating that they both may be important markers of breast cancer progression. Since the metastatic potential of invasive breast cancers is affected by their tumour grade[26], we analysed the levels of *TROLL-2* and *TROLL-3* based on this feature. We found that the levels of both lncRNAs were higher in grade 3 compared to grade 1 and 2 tumours in two distinct TMAs, which included 77 (Biomax TMA) and 154 (Dundee TMA) invasive breast cancers, respectively (Supplementary Fig. 2b–e). Since in the latter TMA the mutational status of *TP53* had been reported[27] and given that mutated forms of p53 are known to inhibit TAp63 functions[3], we analysed the correlation between the presence of mutant p53 and the levels of *TROLL-2* and *TROLL-3*. Notably, we found that both lncRNAs were more abundant in tumours bearing mutant p53 (Supplementary Fig. 2f, g), indicating that *TROLL-2* and *TROLL-3* may be critical breast cancer progression markers in p53 mutant cases. In line with the fact that mutations in *TP53* are more frequent in basal-like tumours (mainly consisting of the so-called triple-negative cases[28]) compared to the other breast cancer subtypes[11], we found that the levels of both lncRNAs were highest in triple-negative breast cancers (TNBCs) in a TMA including 68 breast cancers (Supplementary Fig. 2h, i). Since in this TMA the overall survival data of the patients had been reported[29], we stratified the patients based on the levels of *TROLL-2* and *TROLL-3* and found that high levels of either lncRNA correlated with reduced overall survival of these breast cancer patients (Fig. 2c, d). Together, these data indicate that *TROLL-2* and *TROLL-3* are prognostic factors and may be markers of breast cancer progression.

Given that the expression of *TROLL-2* and *TROLL-3* are elevated in breast tumours and correlate with breast cancer progression, we asked whether *TROLL-2* and *TROLL-3* are required for tumour formation in vivo using two different orthotopic xenograft models of breast cancer, CA1D and MDA MB-231 cells, with the latter growing in vivo due to the inhibition of TAp63 by mutant p53[30,31]. To efficiently reduce the levels of *TROLL-2* and *TROLL-3*, CA1D and MDA MB-231 cells were infected with a doxycycline-inducible shRNA targeting either lncRNA (Supplementary Fig. 2j–m). These cells were then orthotopically injected in the mammary fat pads of nude mice and the mice were fed with doxycycline containing food throughout the experiment. At the end point, we found that the volume of the tumours expressing the shRNAs targeting the lncRNAs were 3–5 times smaller than the tumours derived from the respective control cells (Fig. 2e, f and Supplementary Fig. 2n, o). Next, we assessed the possible role of *TROLL-2* and *TROLL-3* in lung colonization in vivo. To accomplish this, the same cells used for the orthotopic injection experiments were injected in the tail vein of nude mice, and lung colonization was assessed after 5 weeks (MDA MB-231 cells) or 10 weeks (CA1D cells) of doxycycline treatment to downregulate *TROLL-2* and *TROLL-3*. Inspection of the lungs revealed the presence of tumour cells

covering an average area of 7% (CA1D control cells) and 15% (MDA MB-231 control cells) of the lungs (Fig. 2g, h and Supplementary Fig. 2p, q). In contrast, downregulation of either *TROLL-2* or *TROLL-3* severely impaired the lung colonization potential of these breast cancer cells and less than 2% of the lung area was colonised (Fig. 2g, h and Supplementary Fig. 2p, q). The downregulation of *TROLL-2* or *TROLL-3* in the primary tumours was confirmed by ISH staining, which showed a clear reduction in the levels of either lncRNA in the tumours expressing the respective shRNA compared to the strong cytoplasmic signals detected in the control tumours (Supplementary Fig. 2r, s). Together, these data show that downregulation of *TROLL-2* and *TROLL-3* severely impairs the tumorigenic and metastatic potential of two different orthotopic xenograft models of breast cancer and suggest that the breast cancer phenotype observed in a context where TAp63 is inhibited is in part mediated by *TROLL-2* and *TROLL-3*.

**TROLL-2 and TROLL-3 promote cancer progression via WDR26.** LncRNAs are known to affect different molecular processes, including chromatin remodelling, alternative splicing, and miRNA activity, and their effects are achieved by the interaction with specific proteins that ultimately act as their effectors[32,33]. To identify such interacting proteins, we in vitro transcribed the only reported transcript of *TROLL-2* (NR_026825.2) and one of the transcripts of *TROLL-3* (NR_034110.1), which is the isoform that we identified as differentially expressed in the MCF10A breast cancer progression model. These in vitro transcribed RNAs where then used to probe a protein microarray array[34] containing ~9400 human recombinant full-length proteins. This led to the identification of 60 putative interactors of *TROLL-2* and 19 for *TROLL-3*. Seven of these proteins were found as common for both lncRNAs (Fig. 3a, b). Since downregulation of *TROLL-2* and *TROLL-3* showed a similar phenotype in vivo (i.e. reduced formation of mammary tumours and lung colonization), we determined whether any of these 7 common putative interactors could act as effectors of *TROLL-2* and *TROLL-3*. To do this, we over-expressed either *TROLL-2* or *TROLL-3* in CA1D cells and observed an increase in cell migration and invasion in vitro (Fig. 3c, d). Concomitantly, CA1D cells expressing either *TROLL-2* or *TROLL-3* were transfected with siRNAs targeting the 7 identified proteins individually (Supplementary Fig. 3a–i) to establish if the absence of any of them could prevent the increased cell migration and invasion due to *TROLL-2* and *TROLL-3* overexpression. We found that downregulation of two proteins (WDR26 and NCOA5) hindered the migration and invasion potential of CA1D cells overexpressing *TROLL-2* or *TROLL-3* (Fig. 3c, d). A similar but less intense effect was observed in cell proliferation (Supplementary Fig. 3j), while only downregulation of WDR26 showed a modest effect on apoptosis (Supplementary Fig. 3k). Since WDR26 and NCOA5 were required for the oncogenic activities of *TROLL-2* and *TROLL-3*, we further validated the interactions between these two proteins and the two lncRNAs. In vitro transcribed and biotinylated *TROLL-2* and *TROLL-3* efficiently pulled down both WDR26 and NCOA5 (Fig. 3e). These results indicate that both *TROLL-2* and *TROLL-3* interact with WDR26 and NCOA5, which mediate the common pro-oncogenic activities of these two lncRNAs.

**WDR26 localization correlates with breast cancer progression.** Because the expression of *TROLL-2* and *TROLL-3* positively correlate with breast cancer progression, we assessed the expression levels of their 2 interacting proteins, WDR26 and NCOA5, in a TMA of breast cancer progression with 45 samples, comprising normal breast tissue, lobular hyperplasia, DCIS, and

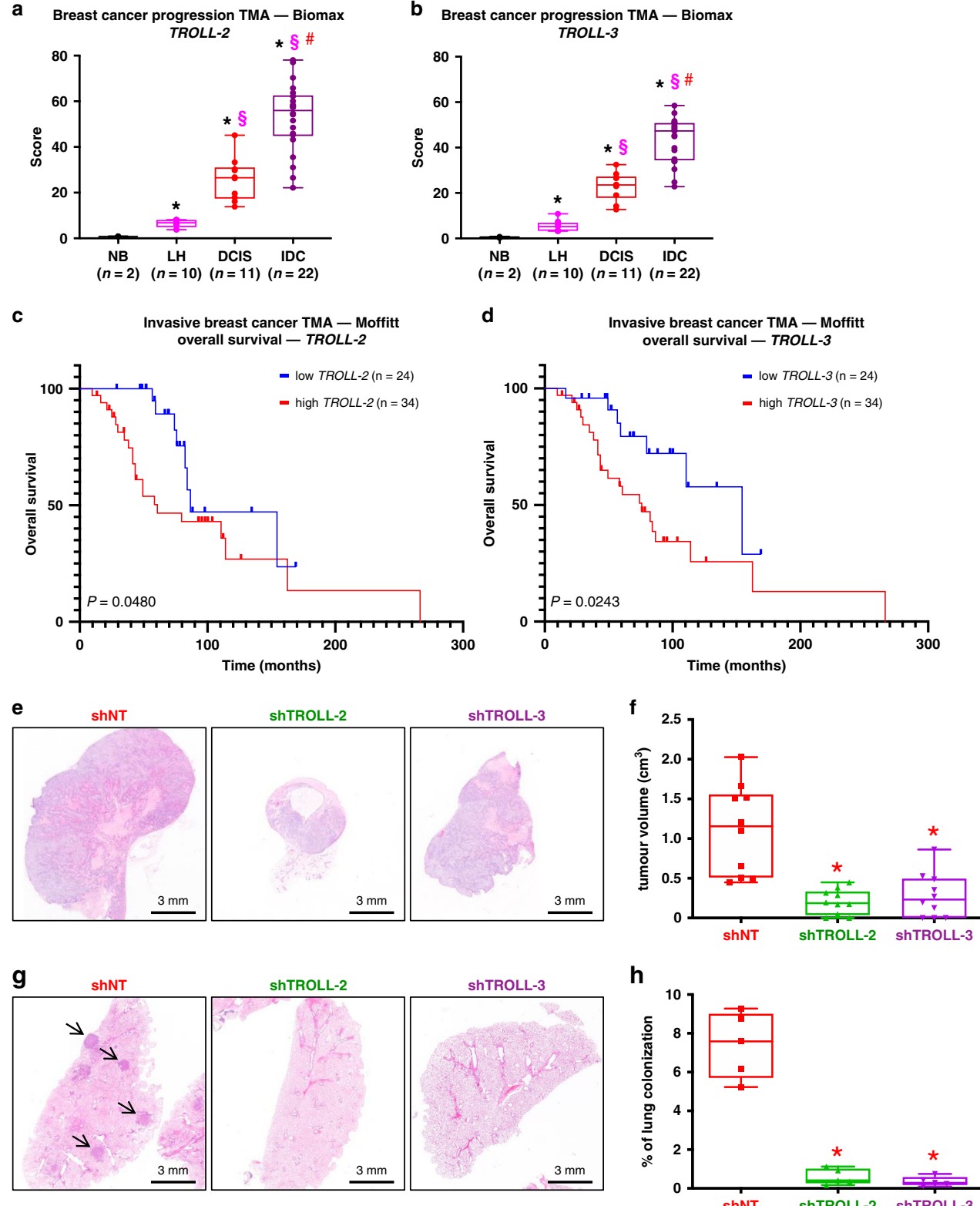

invasive breast cancer biopsies. Interestingly, we found that the cellular localization of WDR26 was mainly nuclear in normal breast tissue and lobular hyperplasia, while in the advanced phases of the disease (i.e. DCIS and invasive ductal carcinoma samples) WDR26 localization was almost exclusively cytoplasmic (Fig. 4a, b). In addition, we found that the expression of both WDR26 and NCOA5 increased over the progression of breast cancer (Supplementary Fig. 4a, b) and were enriched in highly

aggressive and poorly differentiated grade three tumours compared with grade 1 and 2 cases (Supplementary Fig. 4c, d). Importantly, when we examined the localization of WDR26 in the TMA of breast cancer progression based on the ISH score of *TROLL-2* and *TROLL-3*, we found that high levels of these two lncRNAs correlated with high levels and cytoplasmic localization of WDR26 (Fig. 4c, d). This positive correlation was also observed in breast tumours stratified based on tumour grade (Fig. 4e, f).

**Fig. 2 TROLL-2 and TROLL-3 are markers of breast cancer progression. a, b** Quantification of the ISH scores of *TROLL-2* (**a**) and *TROLL-3* (**b**) in the indicated TMA. Data were analysed with two-way ANOVA. Asterisk vs. normal breast tissue (NB), $P < 0.005$. Section sign vs. lobular hyperplasia (LH), $P < 0.005$. Hash vs. ductal carcinoma in situ (DCIS), $P < 0.005$. IDC = invasive ductal carcinoma. **c, d** Kaplan–Meier curves of overall breast cancer survival data based on the levels of *TROLL-2* (**c**) and *TROLL-3* (**d**) in tumours of the indicated TMA. **e** Representative H&E stained cross sections of mammary adenocarcinomas derived from MDA MB-231 cells infected with the indicated shRNAs. **f** Tumour volume quantification of the tumours described in **e**. $n =$ 10 tumours per group, Asterisk vs. shNT (non-targeting), $P < 0.005$, two-tailed Student's $t$ test. **g** Representative H&E stained cross sections of lung colonies derived from CA1D cells infected with the indicated shRNAs. **h** Quantification of the lung colonies described in **g**. $n = 5$ mice for all groups, asterisk vs. NT, $P < 0.005$, two-tailed Student's $t$ test. All boxplots represent the individual data points, median and whiskers (min to max method). Source data are provided as Source data file.

Notably, the higher levels and cytoplasmic localization of WDR26 inversely correlated with the IHC score of TAp63, whose levels are higher in normal breast tissue compared to the other groups, in lines with our previous findings[5] (Supplementary Fig. 4e, f). Together, these data indicate that during breast cancer progression, the levels of TAp63 are reduced while those of *TROLL-2* and *TROLL-3* increase, and this correlates with the cytoplasmic localization of WDR26.

Because the lncRNAs are prognostic factors in breast cancer progression (see Fig. 2c, d) and their levels correlate with those of WDR26, we assessed whether the combination of either lncRNA with WDR26 may be prognostic in breast cancer. To do this, overall TCGA breast cancer survival data[11] were analysed based on the levels of the two lncRNAs and WDR26. While the combination of *TROLL-2* and WDR26 did not reach statistical significance, we found that WDR26 is prognostic in basal-like tumours with high levels of *TROLL-3*, but not with low levels of *TROLL-3* (Supplementary Fig. 4g, h). These findings suggest that when *TROLL-3* levels are high, which correlate with the cytoplasmic localization of WDR26, breast cancers are more aggressive and associated with reduced overall survival.

**Cytoplasmic WDR26 correlates with advanced cancers.** We next asked whether expression of *TROLL-2* and *TROLL-3* correlated with cytoplasmic WDR26 more broadly across other aggressive human cancers by performing a pan-cancer analysis. We performed ISH for *TROLL-2* and *TROLL-3* and IHC for WDR26 in 378 tumour specimens, consisting of 51 ovarian (Biomax TMA, including serous and non-serous adenocarcinomas), 73 colon (Biomax TMA), 55 lung (Biomax TMA, including adenocarcinomas and squamous cell carcinomas), and 199 melanoma cases (Biomax TMA and Moffitt TMA). In line with our observations in breast cancer, all the malignant tumour types assessed had increased levels of *TROLL-2*, *TROLL-3* and WDR26 compared to normal tissue and benign lesions (Fig. 5a). Importantly, increased expression of the two TROLLs and cytoplasmic WDR26 correlated with higher tumour grade (Supplementary Fig. 5a–e'), indicating that the regulation of WDR26 cytoplasmic localization by *TROLL-2* and *TROLL-3* may be a universal mechanism in the progression to aggressive disease.

Given that one of the melanoma TMAs (Moffitt TMA) contains the overall survival data of the patients, we stratified the patients based on the levels of *TROLL-2* and *TROLL-3* and found that high levels of either lncRNA correlated with reduced overall survival of these melanoma patients (Supplementary Fig. 5f', g'). In addition, given that WDR26 is prognostically important in basal-like tumours with high expression of *TROLL-3* (see Supplementary Fig. 4g, h), we verified whether these factors may also be prognostic in other tumour types. To assess this, overall TCGA ovarian cancer[35], colon cancer[36], lung adenocarcinoma[37], lung squamous carcinoma[38], and melanoma[39] survival data were analysed based on the expression levels of *TROLL-3* and WDR26. Among the tested cancer types, we found that WDR26 is prognostic in melanomas with high levels of *TROLL-3*

(Supplementary Fig. 5h', i'). Altogether, our results indicate that the levels of *TROLL-2* and *TROLL-3* and the cytoplasmic location of WDR26 are markers of cancer progression in several human tumour types, and that these factors are prognostic in melanoma.

These results prompted us to verify whether *TROLL-2* and *TROLL-3* are required for the formation and progression of these tumour types in vivo. To accomplish this, we first utilised two orthotopic models of lung adenocarcinoma (H1299 and H358 cells), where *TROLL-2* and *TROLL-3* were downregulated via doxycycline-inducible shRNAs (Supplementary Fig. 5j'–m'). These cells were injected either in the lungs or in the hearts of nude mice to assess their in vivo ability to form primary lung adenocarcinomas[40] or secondary lung colonies[41], respectively. While the control cells successfully developed lung adenocarcinomas (Supplementary Fig. 5n'–q') and lung colonies (Supplementary Fig. 5r'–u'), downregulation of either *TROLL-2* or *TROLL-3* strongly impaired the formation of both (Supplementary Fig. 5n'–u'). The downregulation of *TROLL-2* and *TROLL-3* in the lung adenocarcinomas was confirmed via ISH (Supplementary Fig. 5v'–w'). Next, we utilized two xenograft models of melanoma (A375[42] and Malme-3M[43] cells), which were infected with doxycycline-inducible shRNAs to downregulate either *TROLL-2* or *TROLL-3* (Supplementary Fig. 5x'–a''). The cells were then injected subcutaneously or via tail vein in nude mice to test their ability to generate melanomas and lung colonies, respectively. As observed in both the breast cancer and the lung adenocarcinoma models, downregulation of either lncRNA significantly reduced the tumorigenic (Fig. 5b, c and Supplementary Fig. 5b'', c'') and metastatic (Fig. 5d, e and Supplementary Fig. 5d'', e'') potential of both melanoma cell lines. The effect of the shRNAs in downregulating the respective lncRNA was confirmed by ISH (Supplementary Fig. 5f'', g''). Taken together, these results demonstrate that *TROLL-2* and *TROLL-3* are markers of cancer progression and are necessary for tumour and metastasis formation in multiple cancer types.

**AKT phosphorylation mediates TROLL-2 and TROLL-3's functions.** Given the correlation between high expression of *TROLL-2* and *TROLL-3* and the cytoplasmic localization of WDR26 in invasive human cancers, we hypothesised that these two lncRNAs may promote the cytoplasmic localization of WDR26 to promote metastasis. To test this, we first determined whether WDR26 is localized in the cytoplasm in the MCF10A breast cancer progression model. Indeed, by examining the expression of WDR26 in the nuclear and cytoplasmic fractions of the MCF10A progression model, we found that WDR26 was primarily nuclear in MCF10A cells (representing normal epithelial cells) and cytoplasmic in the metastatic CA1D cells (Supplementary Fig. 6a, b). Further, we transfected CA1D cells with siRNAs targeting *TROLL-2* and *TROLL-3* and assessed WDR26 cellular localization through fractionation. Importantly, we found that downregulation of either lncRNA decreased the pool of WDR26 present in the cytoplasmic fraction promoting its nuclear localization (Fig. 6a and Supplementary Fig. 6c). To

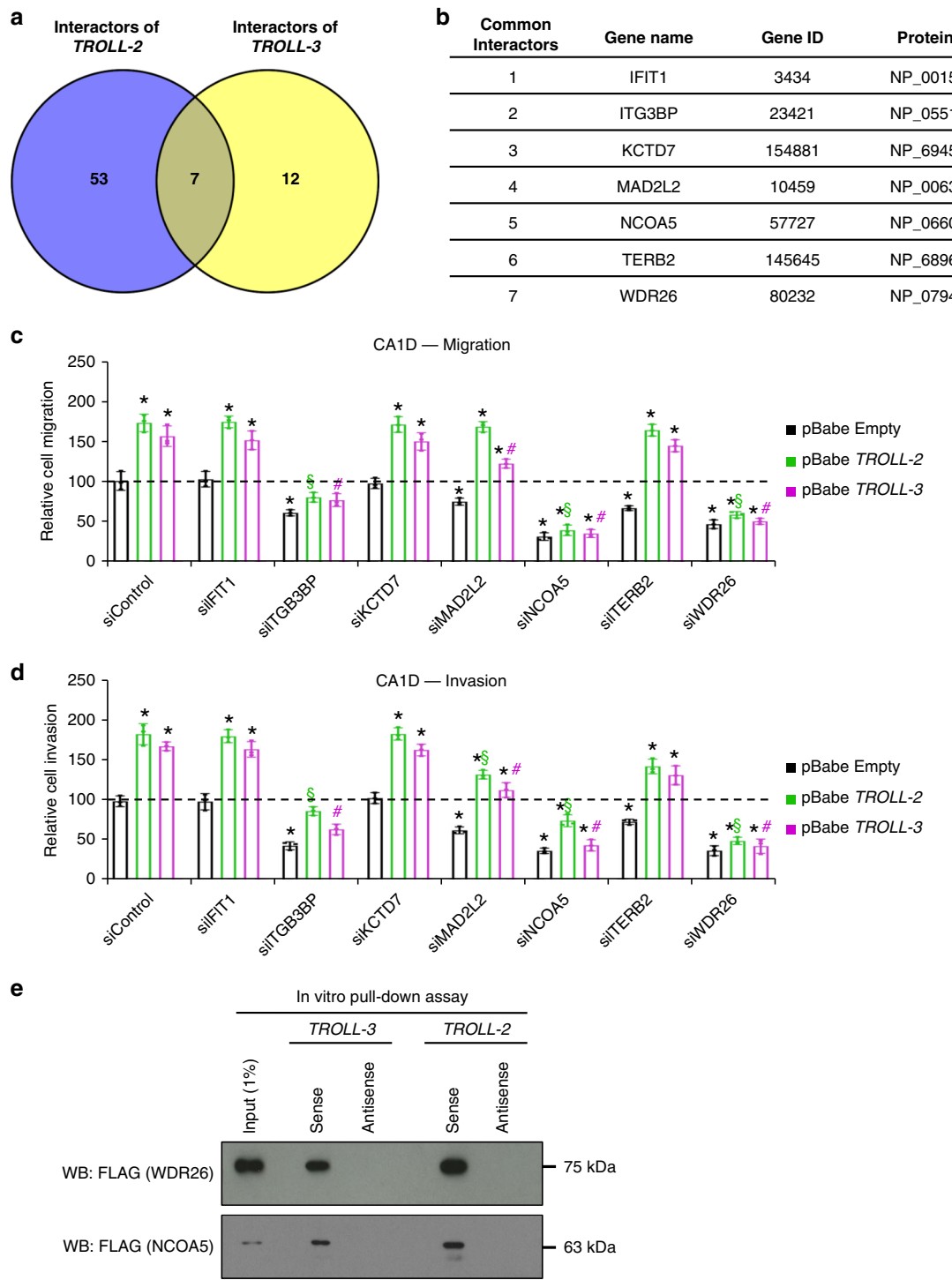

**Fig. 3 TROLL-2 and TROLL-3 promote cancer progression via WDR26. a** Venn diagram of the proteins interacting with *TROLL-2* and *TROLL-3*. **b** Table listing the 7 common interactors of *TROLL-2* and *TROLL-3*. **c, d** Quantification of cell migration (**c**) and invasion (**d**) of CA1D cells overexpressing either *TROLL-2*, *TROLL-3*, or the empty vector, and transfected with the indicated siRNAs. Graphs represent the individual data points, mean ± SD and analysed with two-tailed Student's *t* test, *n* = 3 biological replicates, asterisk vs. pBabe Empty siControl, *P* < 0.005. Section vs. pBabe *TROLL-2* siControl, *P* < 0.005. Hash vs. pBabe *TROLL-3* siControl, *P* < 0.005. **e** Representative western blot for the indicated proteins pulled down by the indicated lncRNAs. *n* = 3 biological replicates. Source data are provided as Source data file.

investigate how WDR26 localization is regulated, we generated WDR26 mutants devoid of either its nuclear localization signal (herein indicated as WDR26-ΔNLS) or its nuclear export signal (herein indicated as WDR26-ΔNES) (Supplementary Fig. 6d) and looked for proteins that differentially interact with these two

mutants in CA1D cells by performing a liquid chromatography-mass spectrometry (LC-MS/MS) analysis. Intriguingly, the top-ranking protein interacting exclusively with WDR26-ΔNES was NOLC1 (also known as Nopp140) (Supplementary Data 1), a shuttling protein[44] known to affect the localization of several

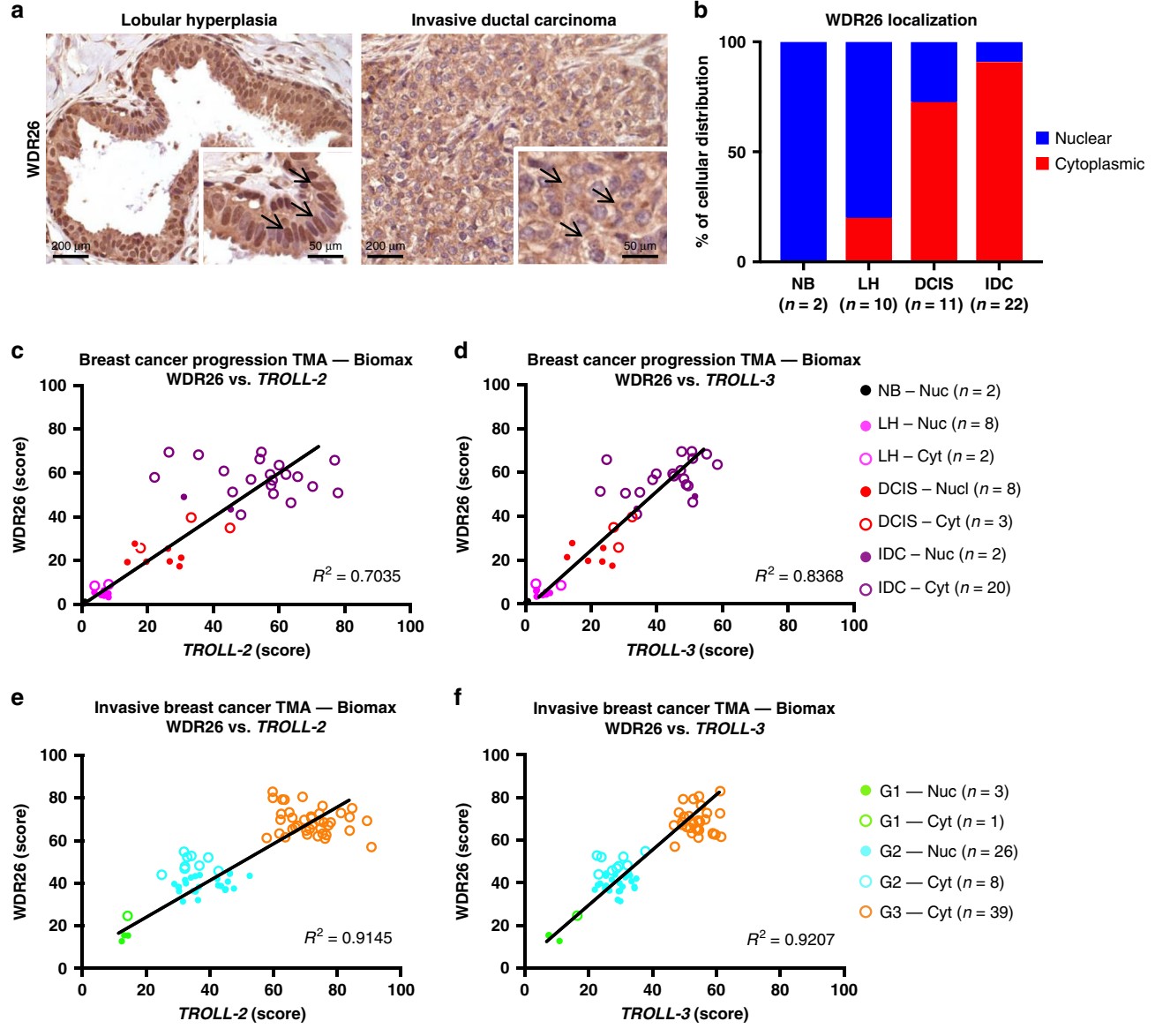

**Fig. 4 WDR26 localization correlates with breast cancer progression. a** Representative IHC images for WDR26 in lobular hyperplasia (left) and invasive ductal carcinoma (right). **b** Quantification of the percentage of WDR26 cellular distribution in the TMA of breast cancer progression shown in Fig. 2a. **c–f** Correlation of the IHC score and cellular distribution of WDR26 with the ISH score of *TROLL-2* and *TROLL-3* in the indicated TMAs based on tissue type (**c**, **d**) or tumour grade (**e**, **f**). Source data are provided as Source data file.

proteins[45–47]. Therefore, we tested whether the interaction between endogenous NOLC1 and WDR26 occurs in the MCF10A progression model and if it is affected by *TROLL-2* and *TROLL-3*. Notably, we found that NOLC1 interacts with WDR26 more efficiently in MCF10A cells, where WDR26 is mainly nuclear and the levels of *TROLL-2* and *TROLL-3* are lower, than in CA1D cells, where WDR26 is mainly cytoplasmic and the levels of *TROLL-2* and *TROLL-3* are higher (Fig. 6b and Supplementary Fig. 6e). Additionally, this interaction is regulated by the two lncRNAs. Indeed, the overexpression of both lncRNAs in MCF10A cells counteracts the binding between NOLC1 and WDR26, while the downregulation of both lncRNAs in CA1D cells promotes it (Fig. 6b and Supplementary Fig. 6e). Since NOLC1 was shown to control the localization of multiple proteins[45–47], we then assessed whether it also affects WDR26 localization. To this aim, we downregulated NOLC1 in MCF10A cells via siRNA and found that reduced levels of NOLC1

increased the pool of WDR26 present in the cytoplasm (Fig. 6c). Together, these data suggest that *TROLL-2* and *TROLL-3* promote WDR26 cytoplasmic localization by preventing its interaction with the shuttling protein NOLC1.

Since reduced expression of *TROLL-2* and *TROLL-3* (see Fig. 1e, f) and of WDR26 (Fig. 3c, d) impair cell migration and invasion of CA1D cells, we tested whether the different cellular distribution of WDR26 can affect the migratory and invasive potential of these cells. To do this, we transfected CA1D cells with an siRNA targeting the 3′UTR of WDR26, thus specifically affecting only endogenous WDR26, and overexpressed WDR26-ΔNLS or WDR26-ΔNES. Notably, the overexpression of WDR26-ΔNLS was able to rescue the cell migration and invasion of WDR26 deficient CA1D cells (Fig. 6d and Supplementary Fig. 6f–k). On the contrary, WDR26-ΔNES showed no appreciable difference in migration or invasion compared to cells treated with empty vector (Fig. 6d and Supplementary Fig. 6f–k),

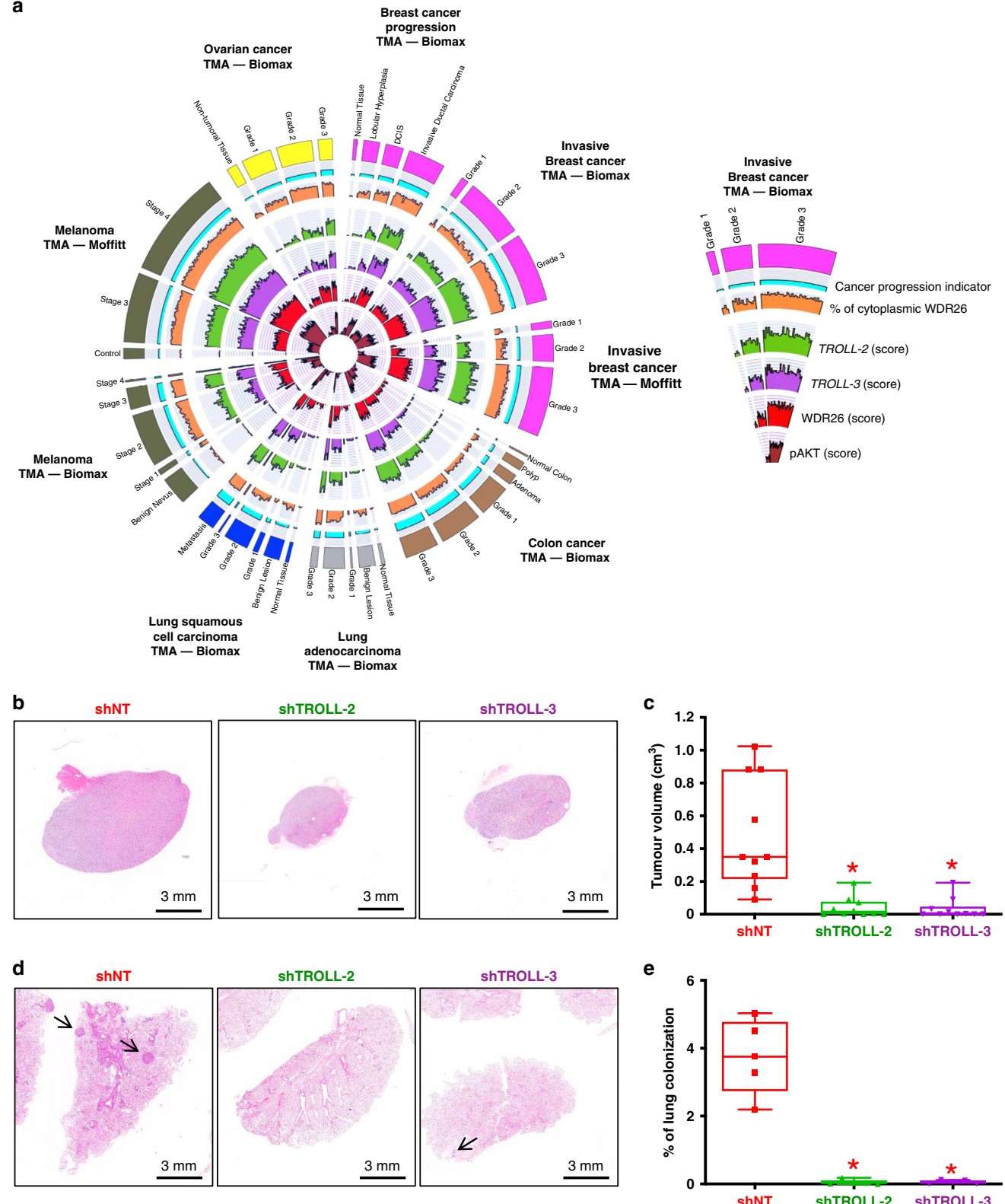

**Fig. 5 Cytoplasmic WDR26 correlates with advanced cancers. a** Circos plot summarizing the expression of *TROLL-2, TROLL-3*, WDR26 and pAKT in TMAs of 378 cancers with progressive disease. **b** Representative H&E stained cross sections of melanomas derived from Malme-3M cells infected with the indicated shRNAs. **c** Quantification of the melanomas described in **b**. $n = 5$ mice for all groups, asterisk vs. NT, $P < 0.005$, two-tailed Student's $t$ test. **d** Representative H&E stained cross sections of lung colonies derived from Malme-3M cells infected with the indicated shRNAs. **e** Quantification of the lung colonies described in **d**. $n = 5$ mice for all groups, asterisk vs. NT, $P < 0.005$, two-tailed Student's $t$ test. Boxplots represent the individual data points, median and whiskers (min to max method). Source data are provided as Source data file.

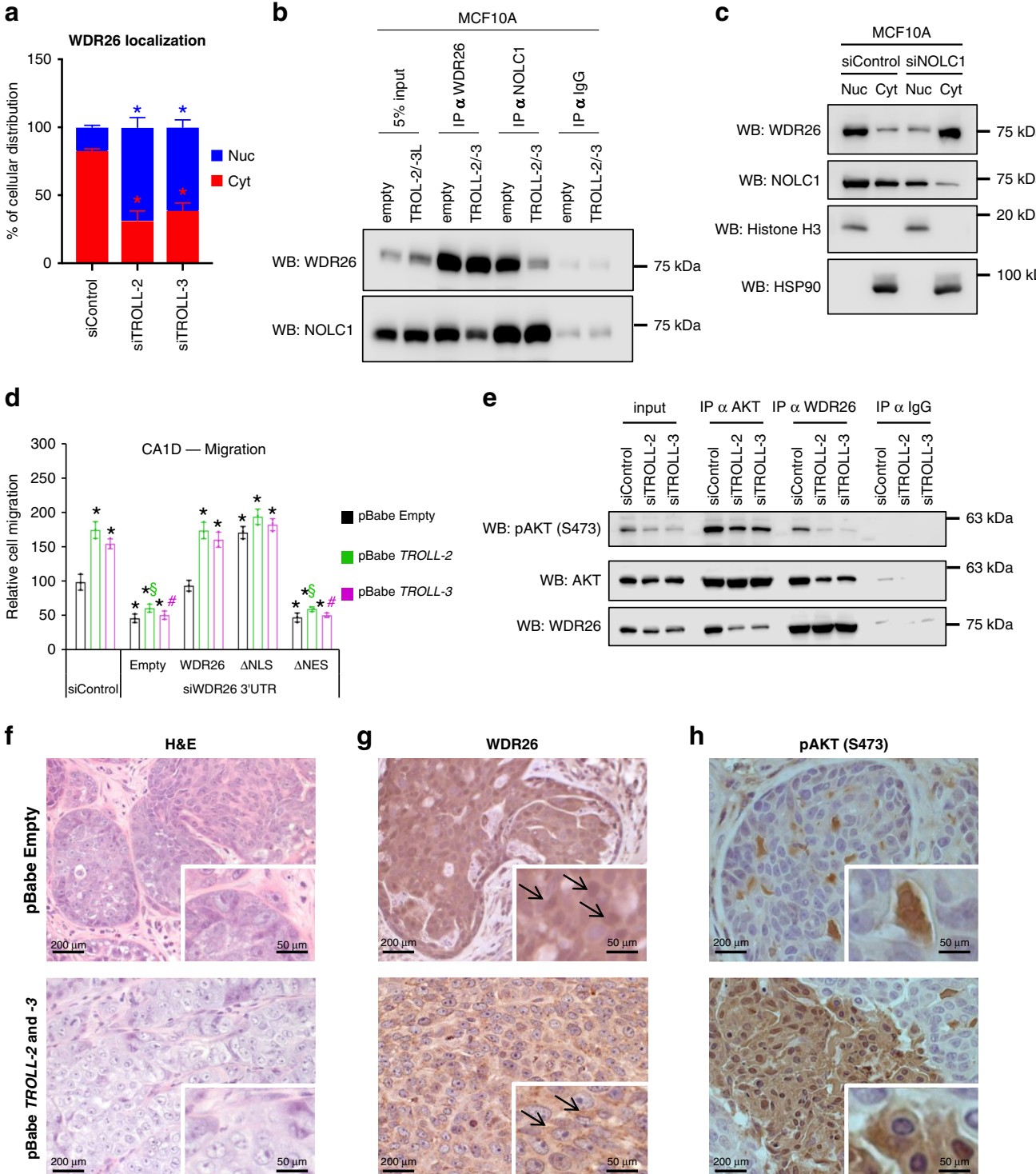

**Fig. 6 AKT phosphorylation mediates *TROLL-2* and *TROLL-3*'s functions. a** Quantification of the percentage of WDR26 localization in the nuclear (Nuc) and cytoplasmic (Cyt) fractions of CA1D cells transfected with the indicated siRNAs. Data are mean ± SD and analysed with two-way ANOVA. $n = 3$ biological replicates, asterisk vs. siControl, $P < 0.05$. **b** Representative western blot analysis of the coimmuno-precipitation of endogenous NOLC1 and WDR26 in MCF10A cells transfected with the indicated constructs and siRNAs. **c** Representative western blot of WDR26 and NOLC1 localization in the nuclear (Nuc) and cytoplasmic (Cyt) fractions of MCF10A cells transfected with the indicated siRNAs. **d** Quantification of cell migration of CA1D cells overexpressing either *TROLL-2*, *TROLL-3*, or empty vector, in combination with the indicated WDR26 constructs and transfected with the indicated siRNAs. Data are mean ± SD and analysed with two-tailed Student's *t* test, $n = 3$ biological replicates, asterisk vs. pBabe Empty siControl, $P < 0.005$. Section vs. pBabe *TROLL-2* siControl, $P < 0.005$. Hash vs. pBabe *TROLL-3* siControl, $P < 0.005$. **e** Representative western blot analysis of the coimmuno-precipitation of endogenous AKT and WDR26 in CA1D cells transfected with the indicated siRNAs. **f–h** Representative H&E images (**f**) and IHC for WDR26 (**g**) and for pAKT (**h**) in tumours derived from DCIS cells infected with the indicated constructs ($n = 5$ mice for both groups). All blots are representative of $n = 3$ biological replicates. Source data are provided as Source data file.

indicating that cytoplasmic WDR26, but not nuclear WDR26, functions downstream of *TROLL-2* and *TROLL-3* and plays a crucial role in cancer progression.

WDR26 is a scaffold protein reported to promote phosphorylation of AKT, a crucial hub of cell survival pathways, and its downstream biological effects[48]. Therefore, we assessed whether cytoplasmic localization of WDR26 had an effect on AKT phosphorylation. To do so, the same CA1D cells expressing WDR26 mutants used for the cell migration and invasion experiments were serum starved for 24 h and AKT phosphorylation triggered via a lysophosphatidic acid (LPA) treatment, known to be WDR26-dependent[48]. Notably, only cytoplasmic WDR26 (i.e. WDR26-ΔNLS) was able to promote AKT phosphorylation to a similar extent as WT WDR26, while nuclear WDR26 (i.e. WDR26-ΔNES) failed to do so (Supplementary Fig. 6l, m), indicating that the cytoplasmic location of WDR26 is required for phosphorylation and induction of AKT signalling.

AKT phosphorylation has been shown to rely on the efficient interaction between PI3K and AKT mediated by WDR26[48]. Since we found that *TROLL-2* and *TROLL-3* control the cellular localization of WDR26 which in turn promotes AKT phosphorylation, we tested whether the two lncRNAs are required for AKT and WDR26 to form complexes. To do this, AKT and WDR26 were individually immunoprecipitated in CA1D cells transfected with siRNAs targeting either lncRNA. Downregulation of either *TROLL-2* or *TROLL-3* strongly impaired the interaction between AKT and WDR26, ultimately leading to a decrease in AKT phosphorylation levels (Fig. 6e). To determine whether AKT phosphorylation is required for the oncogenic activities of *TROLL-2* and *TROLL-3*, we overexpressed either lncRNA in CA1D cells and treated them with the AKT inhibitor MK-2206. The inhibition of AKT strongly impaired the migration and invasion potential of CA1D cells overexpressing *TROLL-2* or *TROLL-3* (Supplementary Fig. 6n–r), indicating that the these two lncRNAs exert their anti-metastatic function through AKT activation.

To investigate the in vivo consequences of *TROLL-2* and *TROLL-3* on WDR26 cellular localization and AKT phosphorylation, we infected the tumorigenic DCIS cells either with an empty vector as a negative control or with both pBabe *TROLL-2* and pBabe *TROLL-3*, and then injected these cells orthotopically in the mammary fat pads of nude mice. Notably, while the DCIS cells gave rise to well-differentiated comedo-like tumours as previously reported[49], DCIS cells overexpressing both lncRNAs produced poorly differentiated tumours (Fig. 6f) reminiscent of invasive ductal carcinomas (IDC). Importantly, this phenotypic change was accompanied both by cytoplasmic localization of WDR26 (Fig. 6g), and by an increased amount of phosphorylated AKT (pAKT) (Fig. 6h). Taken together, these results indicate that *TROLL-2* and *TROLL-3* promote the cytoplasmic localization of WDR26 and its interaction with AKT, which in turn trigger the activation of the AKT pathway to mediate the oncogenic and invasive activities of these two lncRNAs.

Given the correlation between high expression of *TROLL-2* and *TROLL-3*, WDR26 localization, and AKT phosphorylation in DCIS derived tumours, we tested whether such correlation was also present in the breast cancer TMAs that we assessed for the ISH of *TROLL-2* and *TROLL-3* and the IHC of WDR26. Notably, we found that pAKT levels increased with breast cancer progression and tumour grade, and positively correlated with the levels of *TROLL-2* and *TROLL-3* and with the cytoplasmic localization of WDR26 (Fig. 5a and Supplementary Fig. 6s–d′). Next, we used a pan-cancer approach to determine whether *TROLL-2* and *TROLL-3* activated AKT through WDR26 in a broader cancer context. We assessed the correlation between

pAKT levels and either the expression of the two lncRNAs or the localization of WDR26 in TMAs of ovarian cancer, colon cancer, lung adenocarcinoma, lung squamous cell carcinoma, and melanoma. Importantly, in all the tested TMAs we found that high expression of pAKT positively correlated with expression of *TROLL-2* and *TROLL-3* and cytoplasmic expression of WDR26 (Fig. 5a and Supplementary Fig. 6e′-b″), thus indicating that this novel interplay that we have identified is relevant in multiple tumour types.

To characterize this interplay occurring between *TROLL-2*, *TROLL-3*, WDR26, and AKT, we performed a cross-linking immunoprecipitation and qRT-PCR (CLIP-qPCR) assay[50,51] in CA1D cells. We found that not only endogenous WDR26 directly interacts with both lncRNAs (Supplementary Fig. 6c″), but that it is also required for the interaction between these lncRNAs and AKT (Supplementary Fig. 6d″). To map the region in the two lncRNAs necessary for their binding to WDR26, we repeated the CLIP-qPCR assay in the presence of RNAse treatment[50,51]. This approach allowed us to narrow down the portions of the two lncRNAs involved in the interaction with WDR26 to nucleotides 403-627 in *TROLL-2* and nucleotides 360-603 in *TROLL-3* (Supplementary Fig. 6e″, f″). Notably, these two regions contain a common sequence (corresponding to nucleotides 522-538 on *TROLL-2* and 467-482 on *TROLL-3*). To assess if this common sequence is mediating the interaction between the lncRNAs and WDR26, we generated deletion mutants (*TROLL-2* Δ522-538 and *TROLL-3* Δ467-482) and tested their ability to interact with endogenous WDR26 in CA1D cells. Compared to the full-length lncRNAs, deletion of these regions significantly impaired the binding of these lncRNAs to WDR26 (Supplementary Fig. 6g″, h″), thus indicating that endogenous WDR26 binds to *TROLL-2* and *TROLL-3* in CA1D cells and that this interaction is mediated by a common sequence present in both lncRNAs. To determine whether WDR26 and the two lncRNAs form a single complex, we performed a pull-down assay in CA1D cells transfected with siRNA against either lncRNA. Interestingly, we found that reduced levels of either lncRNA impaired the interaction between WDR26 and the remaining lncRNA (Supplementary Fig. 6i″, j″), thus suggesting that both lncRNAs interact at the same time with WDR26 possibly forming a trimeric complex. In line with this hypothesis, we found that the overexpression of one lncRNA cannot rescue the decrease in cell migration and invasion due to the downregulation of the other lncRNA (Supplementary Fig. 6k″, l″).

Taken together, our results demonstrate that *TROLL-2* and *TROLL-3* are markers of cancer progression and regulate the cytoplasmic localization of WDR26 via NOLC1 by forming a trimeric complex with WDR26 that leads to the phosphorylation of AKT and the subsequent induction of oncogenic properties crucial for tumour progression and metastasis formation (Fig. 7).

## Discussion

LncRNAs constitute an ever-growing category of functional RNA species known to impinge on all hallmarks of cancer[33,52,53]. Here, we report the identification of two TAp63-regulated oncogenic lncRNAs or TROLLs, by using both genetically engineered and xenograft mouse models as well as a pan-cancer approach, including the analysis of 723 clinical cases and TCGA overall survival datasets. We found that the expression of two TAp63-regulated lncRNAs, *TROLL-2* and *TROLL-3*, is prognostic in breast cancers and melanomas and positively correlates with the progression of many cancer types, including breast, ovarian, colon, lung adenocarcinoma and lung squamous cell carcinoma, and melanoma. Mechanistically, we demonstrated using xenograft mouse models that *TROLL-2* and *TROLL-3* promote the

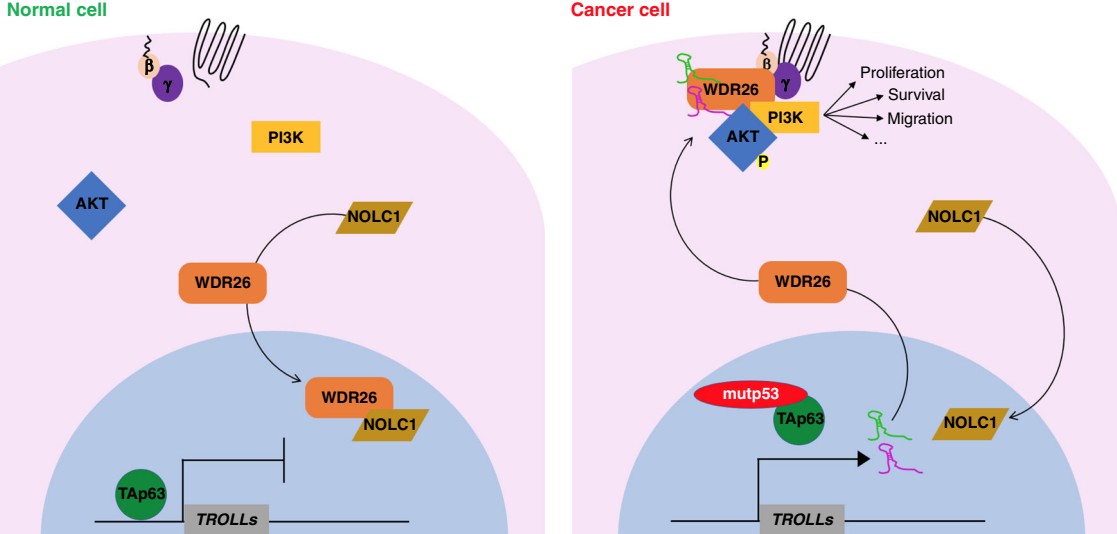

**Fig. 7 Model. Scheme describing the proposed mechanism of action of *TROLL-2* and *TROLL-3*.** In normal cells (e.g. MCF10A cells) the tumour and metastasis suppressor TAp63 inhibits the expression of *TROLL-2* and *TROLL-3*, while NOLC1 interacts with WDR26 and promotes the accumulation of WDR26 in the nucleus. In cancer cells (e.g. CA1D cells), instead, mutant p53 inhibits TAp63, thus allowing for the expression of *TROLL-2* and *TROLL-3*. These lncRNAs counteract the interaction between NOLC1 and WDR26, while promoting the binding of WDR26 to AKT. As a consequence, the PI3K/AKT pathway is activated and can sustain tumour formation and progression.

formation and the progression of different human cancers, including metastatic mammary adenocarcinomas, lung adenocarcinomas, and melanomas. These oncogenic effects are achieved via the scaffold protein WDR26, whose localization is controlled by the two lncRNAs through the shuttling protein NOLC1. In the cytoplasm, the trimeric complex formed by WDR26, *TROLL-2* and *TROLL-3*, leads to the phosphorylation of AKT on Ser473 and the subsequent pro-oncogenic and metastatic effects of the AKT pathway (Fig. 7).

Our mouse-human cross-species analysis, comparing the *TAp63* metastatic mammary adenocarcinoma mouse model[5] and the well characterised MCF10A model of human breast cancer progression[13,14], allowed us to identify a group of 9 TAp63-regulated lncRNAs, which include *TROLL-2* and *TROLL-3*. Both lncRNAs are relevant for metastatic breast cancer. In particular, our observation that the overexpression of *TROLL-2* and *TROLL-3* in DCIS cells is sufficient to promote tumour dedifferentiation and invasive breast cancer indicate that *TROLL-2* and *TROLL-3* are strong drivers of tumour progression. Notably, the levels of both lncRNAs are higher in invasive breast cancers expressing mutant p53, a potent inhibitor of TAp63 function[3], thus making our findings relevant for a large percentage of breast cancer patients (37% of all cases, up to 80% in the basal-like subtype[11]) and possibly for other tumour types harbouring *TP53* mutations.

We have demonstrated that the tumorigenic activities of *TROLL-2* and *TROLL-3* are mediated by one of their common interacting proteins, WDR26, which is a scaffold protein transducing the PI3K signalling pathway[48]. WDR26 contains a WD40 domain, which has been reported to act as a non-canonical RNA binding domain[54,55]. Indeed, several proteins have been shown to interact with lncRNAs via their WD40 domains, as in the case of the association between LRRK2 and *LINK-A*[56] and between LLGL2 and *MAYA*[57]. Thus, we speculate that the WD40 domain of WDR26 may mediate its binding to *TROLL-2* and *TROLL-3*. We have shown that both lncRNAs bind to endogenous WDR26 forming a trimeric complex and that this interaction is mediated by a nucleotide sequence present in both lncRNAs. This complex is important for the localization of WDR26. Indeed, these lncRNAs prevent WDR26 from binding to the shuttling protein

NOLC1 and being sequestered into the nucleus. Instead, the trimeric complex including WDR26 and both lncRNAs localizes in the cytoplasm, where it triggers AKT phosphorylation on Ser473 which is essential to activate the AKT pathway[58]. Importantly, our findings in cellular systems were corroborated not only in orthotopic mouse models but also in tissue microarrays of a broad range of human cancers, indicating that expression of *TROLL-2* and *TROLL-3* represents a common and crucial mechanism of AKT activation in cancer progression. AKT is a pivotal hub funnelling cell growth stimuli and controlling multiple cellular functions, including cell survival, proliferation, and migration[59]. Given the high frequency of oncogenic activating mutations affecting the PI3K/AKT pathway in human tumours[60], our data provide preclinical rationale suggesting that the inactivation of *TROLL-2* and *TROLL-3*, the nuclear sequestration of WDR26, or interfering with the interaction between the TROLLs, WDR26 and AKT might be effective in halting cancer progression and resistance to AKT targeted therapy.

In conclusion, by utilizing mouse models of a broad array of aggressive cancers and by performing a pan-cancer analysis, we have identified biomarkers associated with cancer progression: two TAp63-regulated oncogenic lncRNAs or TROLLs and one of their common interacting proteins, WDR26. The physical and functional interaction of *TROLL-2* and *TROLL-3* with WDR26, as well as its cytoplasmic localization during cancer progression, support the potential use of these factors as markers of cancer progression and as possible predictors of the efficacy for AKT inhibitor-based therapies, and their suitability as therapeutic targets for the development of alternative and, possibly, more effective therapies to treat a broad range of metastatic cancers.

## Methods

**Cell lines and culture conditions**. The MCF10A progression model cell lines (MCF10A, AT1, DCIS, and CA1D) were obtained from the Karmanos Cancer Institute (Detroit, MI) and cultured in Dulbecco's modified Eagle's medium (DMEM)/F12 (1:1) media containing 5% horse serum, 10 μg/mL insulin, 20 ng/mL epidermal growth factor, and 500 ng/mL hydrocortisone. Primary mouse mammary gland epithelial cells were isolated from the 4th pairs of mammary glands of 10-week-old WT and *TAp63*−/− female mice[6], and cultured in DMEM/F12 (1:1) media containing the same components used for MCF10A cells. The human breast

carcinoma cells (MDA MB-231) and lung cancer cells (H1299 and H358) were cultured in DMEM and RPMI media, respectively, containing 10% foetal bovine serum[10,27,61]. The human melanoma cell lines, A375 and Malme-3M, were cultured in DMEM media containing 10 and 20% foetal bovine serum, respectively. All cultured cells were mycoplasma negative.

**Cross-species analysis of coding and non-coding RNAs using RNA sequencing.** The mouse RNA-Seq data[6] was mapped using TopHat[62] against the mouse genome build UCSC mm10 and quantified using Cufflinks[63] against the Gencode[64] mouse gene reference. Data was quantile normalized. We analysed previously published RNA-Seq transcriptomic profiles of the human isogenic MCF10A breast cancer progression model[13], corresponding to normal breast tissue (MCF10A), atypia (AT1), ductal carcinoma in situ (DCIS) and invasive breast cancer (CA1D). The RNA-Seq data was mapped using TopHat[62] against the human genome build UCSC hg19 and quantified using Cufflinks[63] against a combined reference comprised of Gencode[65] and two lncRNA catalogues[66,67]. Using bedtools we separately identified mouse and human pairs of coding/non-coding RNAs within 100 kb from each other. Next, we identified conserved human/mouse genes with a neighbouring non-coding RNA. After this selection process, we obtained 7348 mouse coding genes, 2698 mouse non-coding RNAs, 7770 human coding genes, and 6195 human non-coding RNAs. We used the R package limma[68] to identify differential RNAs for each of the comparisons AT1 over MCF10A, DCIS over AT1, and CA1D over DCIS. We analysed separately the selected 7770 human coding genes, and the 6195 non-coding RNAs, using the fold change cut-off of 1.5x and the FDR-adjusted p-value cut-off of 0.1. We applied the additional constraints that coding genes and non-coding RNAs should be within 100 kb of each other when comparing the same pairs of cells. Finally, we obtained 882 coding and 540 non-coding RNAs. We next considered the corresponding 890 mouse coding genes, and their neighbouring 591 non-coding RNAs. We used limma[68] to analyse RNAs differentially expressed between WT and $TAp63^{-/-}$ MECs. Using the cut-off p-value < 0.05 and fold change exceeding 1.5x, and the genomic distance of at most 100 kb between a coding gene and neighbouring non-coding RNA, we obtained 11 coding genes and 12 non-coding RNAs. Mouse non-coding RNAs were further validated via RT-qPCR, and those that passed, as well as their human counterparts, were depicted graphically as heatmaps using the Python SciPy scientific library.

**Quantitative real time PCR.** Total RNA was prepared using TRIzol reagent (Invitrogen)[5]. For gene expression analysis, complementary DNA was synthesized from 5 μg of total RNA using the SuperScript II First-Strand Synthesis Kit (Invitrogen) according to the manufacturer's protocol followed by qRT-PCR using the TaqMan® Universal PCR Master Mix (Applied Biosystems). qRT-PCR was performed using a QuantStudio 6 flex PCR machine (Applied Biosystems) and each qRT-PCR was performed in triplicate. The utilized primers are listed in Supplementary Table 1.

**Plasmids, siRNAs and shRNAs.** pBabe-RPSAP52 (*TROLL-2*) was generated by subcloning RPSAP52 from pBluescript II SK hRPSAP52 (NR_026825.2, Dharmancon) into pBabe-hygro (Addgene). pBluescript II SK TROLL-2 Δ522-538 was generated via deletion of the indicated nucleotides from pBluescript II SK hRPSAP52. pBabe-TRAF3IP2-AS1 (*TROLL-3*) was generated by subcloning TRAF3IP2-AS1 from pCMV-SPORT6 hTRAF3IP2-AS1 (NR_034110.1, Dharmacon) into pBabe-hygro (Addgene). pCMV-SPORT6 TROLL-3 Δ467-482 was generated via deletion of the indicated nucleotides from pCMV-SPORT6 hTRAF3IP2-AS1. For DNA transfection Lipofectamine 2000 (Invitrogen) was used and for siRNA transfection double-stranded RNA oligos (40 nM) were transfected using Lipofectamine RNAiMax (Invitrogen) according to the manufacturer's instructions. The universal negative siRNA control #1 (siControl) was purchased from Sigma (SIC001-10NMOL). Additional siRNAs utilized were: siIFIT1 (SASI_Hs01_00017406, Sigma), siITG3BP (SASI_Hs01_00238825, Sigma), siKCTD7 (SASI_Hs01_00228145, Sigma), siMAD2L2 (SASI_Hs02_00329127, Sigma), siNCOA5 (SASI_Hs01_00172441, Sigma), siTERB2 (SASI_Hs01_00102225, Sigma), siNOLC1 (SASI_Hs01_00116300, Sigma), siWDR26 (SASI_Hs01_00029068, Sigma), and siWDR26 3'UTR (5'–UGAUAGAAAGAGUGCAUUA–3'). The sequences of the siRNA pools used to target the lncRNAs are listed in Supplementary Table 2. The shRNA targeting *TROLL-2* (based on siRNA#2 5'-GAAGGAAUUUCAGGGUAAA-3') and the shRNA targeting TROLL-3 (based on siRNA#3 5'- GCTATGCAGGATTGGCAGG-3') were designed and cloned into pLV-H1TetO-GFP-Puro lentiviral vector (SORT-CO1, Biosettia) in accordance with the manufacturer's protocol. The provided non-targeting shRNA (shNT) was used as a negative control. The generated doxycycline-inducible constructs were then utilized to infect the cell lines of interest with virus-containing media supplemented with 2 μg/mL polybrene for 24 h. Cells were then selected for 48 h with puromycin (2 μg/ml)[61].

**Chromatin immunoprecipitation.** CA1D cells were either treated for 6 days with doxycycline (1 μg/mL) or left untreated. Cellular proteins were crosslinked to DNA using 1% formaldehyde and chromatin was isolated and sonicated in ChIP buffer (20 mM Tris-HCl pH 7.5, 100 mM NaCl, 1 mM EDTA, 0.5% NP40, 0.5% sodium deoxycholate, 0.1% SDS)[10]. Each ChIP was performed in triplicate using

either 2 μg of a TAp63 specific antibody (sc-8608, Santa Cruz) or IgG purified from mouse serum (sc-2025, Santa Cruz) and rabbit serum (sc-2027, Santa Cruz) as a negative control for the immunoprecipitation. The recruitment of TAp63 was analysed by qRT-PCR comparing regions with a TAp63 binding sites (TAp63 BS) vs. non-specific regions (NS BS) with the primers listed in Supplementary Table 3.

**Cell proliferation and apoptosis assays.** Cells were plated at a density of $1 \times 10^4$ cells in 6 replicates in a 96-well plate. Three biological replicates were performed per each assay. To evaluate cell proliferation, the cells were labelled for 3 h with 10 mM EdU (5'-ethynyl-2'-deoxyuridine) and stained using the Click-iT EdU microplate assay (Invitrogen). Apoptosis was monitored by incubating the cells with Annexin V-Alexa Fluor 488 (Essen BioScience) according to the manufacturer's instructions. Images were captured and percent of either cell proliferation or apoptosis was quantified using a high-throughput immunofluorescence plate reader and accompanying software (IncuCyte, Essen Bioscience).

**Cell migration and invasion assays.** In all, $1 \times 10^4$ cells in DMEM/F12 (1:1) media containing 0.5% horse serum were plated in 6 replicates in an IncuCyte ClearView 96-well cell migration plate (Essen BioScience), whose wells were either left uncoated (cell migration) or coated with 20 μL of 200 μg/mL growth factor reduced matrigel (Corning) (cell invasion). If indicated, the cells were treated for the duration of the assay with 100 nM MK-2206 (S1078, Selleckchem). As chemo-attractant, DMEM/F12 (1:1) media containing 5% horse serum, 10 μg/mL insulin, 20 ng/mL epidermal growth factor, and 500 ng/mL hydrocortisone was used in the bottom chambers. Images were captured and percent of either cell migration or invasion was quantified using a high-throughput plate reader and accompanying software (IncuCyte, Essen Bioscience).

**Orthotopic xenograft mouse model.** Female athymic *nu/nu* mice (6 weeks old) were used for all the experiments and randomized into three groups of 5 mice each: (i) cells infected with shRNA control (shNT); (ii) cells infected with shRNA for *TROLL-2* (shTROLL-2); and (iii) cells infected with shRNA for *TROLL-3* (shTROLL-3). For the experiments involving the breast cancer cell lines, $2 \times 10^6$ cells (CA1D) or $2.5 \times 10^6$ cells (MDA MB-231) in 100 μL of growth factor reduced matrigel (Corning) were implanted orthotopically into the 4th pair of mammary fat pads. For the experiments involving the lung cancer cell lines, $1 \times 10^6$ cells (H1299) or $2 \times 10^6$ cells (H358) in 100 μL of PBS were delivered percutaneously into the left lateral thorax, at the lateral dorsal axillary line, ~1.5 cm above the lower rib line just below the upper border of the scapula[40]. For the experiments involving the melanoma cell lines (A375 and Malme-3M), $1 \times 10^7$ cells in 100 μL of PBS were subcutaneously injected in both flanks. Mice were fed with doxycycline containing food (200 mg/kg) to induce the expression of the shRNA and target the lncRNA of interest for the entire duration of the experiment, which was either 5 weeks (MDA MB-231), 6 weeks (H1299, A375, and Malme-3M), 8 weeks (H358), or 10 weeks (CA1D). At the indicated end point, the tumour xenografts were collected, measured with a calliper, and analysed using ISH. For the experiments involving DCIS cells, female athymic *nu/nu* mice (6 weeks old) were randomized into two groups of 5 mice each: DCIS infected with pBabe Empty and DCIS infected with both pBabe *TROLL-2* and pBabe *TROLL-3*. The obtained tumour xenografts were collected 5 weeks after the injection, measured with a calliper, and analysed using IHC. All procedures were approved by the IACUC at the H. Lee Moffitt Cancer Center and Research Institute.

**Tail vein injections.** Female athymic *nu/nu* mice (6 weeks old) were randomized into three groups of 5 mice each as described above for the orthotopic injections. The following amounts of cells in 100 μL of PBS were injected in the tail vein of the mice: $5 \times 10^5$ cells (CA1D), $1 \times 10^6$ cells (MDA MB-231), $5 \times 10^6$ cells (A375 and Malme-3M). Mice were fed with doxycycline containing food (200 mg/kg) to induce the expression of the shRNA and target the lncRNA of interest throughout the duration of the experiment, which was either 4 weeks (MDA MB-231), 8 weeks (A375 and Malme-3M), or 10 weeks (CA1D). At the indicated end point, the lungs were collected and fixed in buffered formalin. Hematoxylin and eosin (H&E) stained cross sections were then used to quantify the area of the lungs colonized by the cancer cells via the Oncotopix® software (Visiopharm). All procedures were approved by the IACUC at the H. Lee Moffitt Cancer Center and Research Institute.

**Intracardiac injections.** Female athymic *nu/nu* mice (6 weeks old) were randomized into three groups of 5 mice each as described above for the orthotopic injections. In all, $8 \times 10^5$ cells (H1299) and $2 \times 10^6$ cells (H1299) in 100 μL of PBS were injected in the left cardiac ventricle[41]. Mice were fed with doxycycline containing food (200 mg/kg) to induce the expression of the shRNA and target the lncRNA of interest. 4 weeks after the injection, the lungs were collected and fixed in buffered formalin. Hematoxylin and eosin (H&E) stained cross sections were then used to quantify the area of the lungs colonized by the cancer cells via the Oncotopix® software (Visiopharm). All procedures were approved by the IACUC at the H. Lee Moffitt Cancer Center and Research Institute.

**Moffitt tissue microarrays**. The Moffitt tissue microarrays (TMAs) used in this study consisted of: (i) 68 breast cancer samples comprising 43 triple negative and 25 non-triple negative breast cancers (TMA-5)[29]; and (ii) 100 melanoma samples and 6 control cases (TMA-4). The formalin-fixed and paraffin-embedded biopsies were used to produce 0.6 mm cores, which were assembled into the two TMAs by the Tissue Core Facility at the H. Lee Moffitt Cancer Center & Research Institute under delegated ethical authority of the Moffitt Research Ethics Committee with written informed consent from contributing patients.

**In situ hybridization of xenograft tumours and tissue microarrays**. The xenograft tumours, TMAs of breast cancer progression (BR480a, US Biomax), colon cancer progression (CO961, US Biomax), lung cancer progression (BC04002a, US Biomax), ovarian cancer progression (OV1005b, US Biomax), two TMAs of melanoma progression (ME1004f, US Biomax; and the Moffitt TMA-4, Moffitt Cancer Center), and three TMAs of invasive breast cancers (BR20837a, US Biomax; the Dundee TMA[27], Tayside Tissue Bank; and the Moffitt TMA-5, Moffitt Cancer Center) were used for the ISH assay. The double digoxigenin labelled LNA probes (Exiqon) utilized for ISH were:
_ *TROLL-2* (5′-ACAGAAGCTTGCAGGGAACCT-3′);
_ *TROLL-3* (5′-TCGGCGAGGCAAGTGTGAGCA-3′).
As a negative control, the double digoxigenin labelled scramble LNA probe (339508, Exiqon) was used.

The ISH was performed using the Exiqon protocol for FFPE tissue, and the hybridization step was done using a 150 nM final concentration of the LNA probes at 55 °C for 1 h in the Dako hybridizer (Agilent). The LNA probes were then detected with Alkaline Phosphatase (AP) conjugated antibody (11093274910, Sigma, 1:400), and visualized via a chromogenic reaction converting the AP substrate NBT-BCIP (11697471001, Roche) into an alcohol insoluble purple precipitate. Nuclear Fast Red™ (H-3403, Vector laboratories) was used as a counterstain. The signal intensity (continuous variable, 0–1) and the proportion of positive tissue (continuous variable, 0–100%) were measured using the Oncotopix® software (Visiopharm). The ISH score was then quantified by multiplying the signal intensity by the proportion of positive tissue, giving a value comprised between 0 and 100, and visualized using the Circos software[69].

**Immunohistochemistry of xenograft tumours and tissue microarrays**. CA1D-derived xenograft tumours, TMAs of breast cancer progression (BR480a, US Biomax), colon cancer progression (CO961, US Biomax), lung cancer progression (BC04002a, US Biomax), ovarian cancer progression (OV1005b, US Biomax), two TMAs of melanoma progression (ME1004f, US Biomax; and the Moffitt TMA-4, Moffitt Cancer Center), and three TMAs of invasive breast cancers (BR20837a, US Biomax; the Dundee TMA[27], Tayside Tissue Bank; and the Moffitt TMA-5, Moffitt Cancer Center) were used for immunohistochemistry (IHC). IHC was performed overnight keeping the slides in humified chambers[61], and incubating them with the following primary antibodies: NCOA5 (ab70831, Abcam, 1:200), WDR26 (ab203345, Abcam, 1:200), TAp63 (618902, Biolegend, 1:200) and pAKT (S473) (4060S, Cell Signaling, 1:100). In the case of the TMAs, the signal intensity (continuous variable, 0 to 1) and the proportion of positive tissue (continuous variable, 0–100%) were measured using the Oncotopix® software (Visiopharm). The IHC score was then quantified by multiplying the signal intensity by the proportion of positive tissue, giving a value comprised between 0 and 100, and visualized using the Circos software[69].

**Cy5 labelling of lncRNAs and protein microarray analysis**. In vitro transcription of the sense and antisense strands of *RPSAP52* (*TROLL-2*), *TRAF3IP2-AS1* (*TROLL-2*), and their respective deletion mutants (*TROLL-2* Δ522-538 and *TROLL-3* Δ467-482) was performed from pBluescript II SK and pCMV-SPORT6, respectively. Following transcription, the strands were labelled with Cy5 using the Label IT μArray Cy5 labelling kit (Mirus) with a labelling efficiency of 3 pmol Cy5 dye per μg of RNA. Protoarray Human Protein Microarrays v5.0 (ThermoFisher Scientific) were used for the hybridization with 10 pmol of either Cy5 labelled sense or antisense as a negative control. Three independent replicates were carried out with the hybridization step performed at 4 °C for 1 h in the dark[34]. The slides were then scanned with the GenePix 4000B Microarray (Molecular Devices) at 635 nm for the selection of proteins interacting exclusively with the sense strands.

**In vitro RNA pull-down coupled with protein detection**. For the in vitro RNA pull-down, the magnetic RNA-protein pull-down kit (Pierce) was used according to the manufacturer's instructions. Briefly, in vitro transcribed lncRNAs were end-labelled with desthiobiotin. In all, 50 pmol of labelled lncRNA was incubated with 50 μl streptavidin magnetic beads for 30 min at 25 °C with agitation. Streptavidin magnetic bead-bound lncRNA was then incubated with cell lysate (33-330 μg) of either CA1D cells (for endogenous WDR26), CA1D cells overexpressing FLAG-tagged WDR26 (OHu01176D, GenScript), or HEK293T cells overexpressing FLAG-tagged NCOA5 (OHu03595D, GenScript). After an overnight incubation at 4 °C with gentle end-to-end rotation, the beads were washed three times with 1X Wash Buffer provided in the kit. After the final wash, streptavidin magnetic beads were resuspended in 50 μl of Elution Buffer provided in the kit, and the eluted RNA-bound proteins were analysed by SDS-PAGE[61]. The detection of FLAG-

tagged WDR26 and FLAG-tagged NCOA5 was performed with the anti-Flag antibody (A8592, Sigma, 1:1000). The detection of endogenous WDR26 was performed with the anti-WDR26 antibody (ab85961, Abcam, 1:2000).

**Nuclear and cytoplasmic protein fractionation**. In all, $3 \times 10^6$ cells were used for the extraction of nuclear and cytoplasmic proteins with the subcellular protein fractionation kit for cultured cells (78840, ThermoFisher Scientific) in accordance to the manufacturer's instructions. In all, 10 μg of the nuclear and cytoplasmic fractions were then analysed by SDS-PAGE[61], and the following primary antibodies were used: WDR26 (ab85961, Abcam, 1:2000), NOLC1 (sc-374033, Santa Cruz, 1:1000), FLAG (A8592, Sigma, 1:1000), H3 (ab1791, Abcam, 1:2000), and HSP90 (ab13495, Abcam, 1:5000).

**Kaplan–Meier curves of TCGA data**. To assess the clinical significance of composite protein coding genes (PCG) and lncRNAs we first downloaded the TCGA breast cancer[11] and melanoma[39] data transcriptome profiles using the Firebrowse[70] portal. For a combination of PCG and lncRNA, the samples were first separated into two equal bins with respect to each RNA expression. Resulting groupings from PCG and lncRNA were combined so each sample will belong to one of the four groups. The four groups were: (1) PCG expression > median, lncRNA expression > median; (2) PCG expression > median, lncRNA expression < median; (3) PCG expression < median, lncRNA expression > median; and (4) PCG expression < median, lncRNA expression < median. Groups association with survival was assessed using the survival package[71] in the R statistical system.

**WDR26 deletion mutants**. The amino acid sequence of WDR26 was analysed for the presence of a nuclear localization signal (NLS) with cNLS Mapper[72], and of a nuclear export signal (NES) with NetNES[73]. The identified NLS was between aa 111 and aa 121 (GSSLKKKKRLS), while the NES was localized between aa 224 and aa 236 (LEDGKVLEEALQVL). These sequences were deleted from pcDNA3.1-WDR26 FLAG (OHu01176D, GenScript) to produce WDR26-ΔNLS and WDR26-ΔNES, respectively.

**Liquid chromatography-mass spectrometry (LC-MS/MS) analysis**. In all, $1 \times 10^6$ CA1D were transfected with either WDR26 FLAG, WDR26-ΔNLS FLAG, WDR26-ΔNES, or pcDNA3.1-FLAG as a negative control. 24 h after the transfection, the cells were lysed and the IP assay was performed in 50 mM Tris-HCl pH 8.0, 150 mM NaCl, 1 mM EDTA, 1% NP-40[27] and using 25 μl of the anti-FLAG M2 magnetic beads (M8823, Sigma). The immunoprecipitated proteins were extracted, digested with trypsin, and analysed via LC-MS/MS[74] and the identified peptides are listed in Supplementary Data 1.

**Co-immunoprecipitation (CoIP) assay**. In all, $1 \times 10^7$ CA1D were transfected with siRNAs for *TROLL-2*, *TROLL-3*, or with the non-targeting siRNA as a negative control. To test the interaction between WDR26 and AKT, 24 h after the transfection the cells were serum-starved for 24 h and subsequently treated for 10 min with 10 μM lysophosphatidic acid (LPA). To test the interaction between WDR26 and NOLC1, the cells were kept in their growth media for 48 h. The cells were then lysed and the CoIP assay was performed in 50 mM Tris-HCl pH 8.0, 150 mM NaCl, 1 mM EDTA, 1% NP-40[27], and 1 μg of each of following primary antibodies was utilized per sample: AKT (9272 S, Cell Signaling), NOLC1 (ab184550, Abcam), WDR26 (ab203345, Abcam), and normal rabbit IgG (sc-2027, Santa Cruz) as negative control. The interaction was then detected via western blot using the following primary antibodies: pAKT (S473) (4060 S, Cell Signaling, 1:100), AKT (ab8805, Abcam, 1:1000), and WDR26 (ab85961, Abcam, 1:2000).

**LPA treatment**. In all, $1 \times 10^6$ CA1D cells were transfected either with the siControl or with the siWDR26 3′UTR in combination with either pcDNA3.1-FLAG, pcDNA3.1-WDR26 FLAG, pcDNA3.1-WDR26-ΔNLS FLAG, or pcDNA3.1-WDR26-ΔNES FLAG. In all, 24 h after the transfection, the cells were serum-starved for 24 h and subsequently treated for 10 min with 10 μM lysophosphatidic acid (LPA). The cells were then lysed and western blot was performed[61] and the following primary antibodies were used: pAKT (S473) (4060 S, Cell Signaling, 1:100), AKT (ab8805, Abcam, 1:1000), WDR26 (ab85961, Abcam, 1:2000), FLAG (A8592, Sigma, 1:1000), and Actin (A5441, Sigma, 1:5000).

**Cross-linking immunoprecipitation and qRT-PCR (CLIP-qPCR) assay**. To test the interaction between the lncRNAs and either AKT or WDR26, $1 \times 10^7$ CA1D were transfected with siRNAs for WDR26 or with the non-targeting siRNA as a negative control. In all, 48 h after the transfection, the cells were incubated overnight with 100 μM 4-thiouridine to allow for the UV-driven crosslinking between RNAs and interacting proteins. The cells were then lysed in 20 mM Tris–HCl at pH 7.5, 100 mM KCl, 5 mM MgCl2, and 0.5% NP-40[50,51] and the proteins of interest were immunoprecipitated using 1 μg of each of the following primary antibodies: AKT (9272 S, Cell Signaling), WDR26 (ab203345, Abcam), and normal rabbit IgG (sc-2027, Santa Cruz) as negative control. The presence of the lncRNAs in the CLIP-ed samples was assessed via qRT-PCR using the Taqman probes listed in Supplementary Table 1. To map the regions of *TROLL-2* and *TROLL-3* interacting

with WDR26, $1 \times 10^7$ CA1D were utilized. The CLIP assay was performed as above and including an RNAse T1 treatment step (1 U/μL at 22 C for 10 min)[50,51] and using 1 μg of each of the following primary antibodies: WDR26 (ab203345, Abcam) and normal rabbit IgG (sc-2027, Santa Cruz) as negative control. The presence of the lncRNA fragments in the CLIP-ed samples was assessed via qRT-PCR using the primers listed in Supplementary Table 5.

**Statistical analyses and reproducibility**. All the experiments are representative of at least three independent replicates. Data collection was performed with Microsoft Excel and data analysis was performed using GraphPad Prism, Oncotopix® (Visiopharm), and R software. Gels, blots and images are representative of three independent experiments giving similar results.

**Reporting summary**. Further information on research design is available in the Nature Research Reporting Summary linked to this article.

## Data availability

Public available data used in this paper were obtained from UCSC (http://genome.ucsc.edu/cgibin/hgGateway) and The Cancer Genome Atlas (TCGA) breast cancer and melanoma datasets (http://www.cancer.gov/about-nci/organization/ccg/research/structural-genomics/tcga). The list of peptides interacting with WDR26 and its mutants is provided as Supplementary Data 1. All the other data (imaging) supporting the findings of this study are available from the corresponding author upon reasonable request. Source data are provided with this paper.

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

## Acknowledgements

We thank Joseph Johnson and his team at the Analytic Microscopy Core Facility (Moffitt Cancer Center) for the quantification of the IHC and ISH signals, Mahmoud A. Abdalah and Olya Stringfield at the Quantitative Imaging Core (Moffitt Cancer Center) for the quantification of the lung metastases, and Lancia N. Darville-Bowleg and John M. Koomen at the Proteomics and Metabolomics Core (Moffitt Cancer Center) for the LC-MS/MS analysis. E.R.F. is a National Cancer Institute Outstanding Investigator (R35CA197452), Moffitt Distinguished Scholar, and Scholar of the Leukemia and Lymphoma Society, the Rita Allen Foundation, and the V Foundation for Cancer Research. M.N. is a Scholar of the Cancer Prevention Research Institute of Texas-Translational Research in Multidisciplinary Program and was supported by a Research Training Award (RP140106) and a grant from the Cancer Prevention and Research Institute of Texas (RP150094). K.R. and C.C. have been supported in part by NIH P30 shared resource grant CA125123, CPRIT RP170005, and NIEHS P30 Center grant 1P30ES030285. We gratefully acknowledge the assistance of Jane L. Messina (Moffitt Cancer Center) in determining the suitability of melanoma TMAs generated at Moffitt Cancer Center and the assistance of Carolyn Rich (Moffitt Cancer Center) in procuring the appropriate TMA sections. This work has been supported in part by the Tissue Core (Research Histology), Analytic Microscopy Core, Image Response Assessment Team, and Proteomics Core Facilities at the H. Lee Moffitt Cancer Center & Research Institute (P30-CA076292).

## Author contributions

M.N. and E.R.F. conceived the study, designed experiments and analysed data. M.N., X.L., H.D.A., A.A.D., I.Barannikov and M.A.P. designed and performed experiments. M.N., X.L., A.T., P.Q., C.C., K.Y.T. and E.R.F. analysed the data. I. Bedrosian provided the RNA-sequencing data of the MCF10A breast cancer progression model. A.T. provided the Dundee TMA. J.M. and P.Q. performed statistical analyses on the Dundee TMA data. K.R., C.C. and P.H.G. performed bioinformatic analyses. D.C.M. and A.M.M. provided the Moffitt TMA-5. E.R.F. and M.N. wrote the paper. All authors discussed the paper and commented on the paper.

## Competing interests

The authors declare no competing interests.
