## [Peer Review File · Nature Communications]

Reviewers' comments:

Reviewer #1 (Remarks to the Author):

This study characterizes TAp63 regulated lncRNAs (TROLLs). P63 is a p53 ortholog plays an important role in the suppression of tumor metastasis and invasion. By using locus conservation approach, authors have chosen 8 lncRNAs (TROLLs) out of 54 differentially expressed lncRNAs. For further functional investigations, 2 lncRNAs, TROLL-2/-3, have been chosen based on consensus TAp63 bindings sites in their promoter regions and ChIP-qPCR data. TROLLs expression correlates breast cancer progression, and their higher expression correlates with cytoplasmic localization of their common interacting protein WDR26 and activation of pro-survival pathways PI3K/AKT pathways.

Overall, this is an interesting study provide insights into lncRNA dependent gene regulatory mechanisms underlying the tumor progression. I have several concerns listed below.

Major comments

Figure 1A and 1E: Recently, MALAT1 has recently been shown to suppress breast cancer by attenuating the function of metastatic promoting factor TEAD (Kim J et al., 2018 Dec. Nature Genetics). However, in the current study TAp63 KD results in the activation of MALAT1, and increased cell migration/invasion. How do these two contrasting observations on MALAT1 reconcile?

Most of these lncRNA already have names and why they should renamed with TROLLs. It only confuses the research community with several aliases for a single lncRNA. Moreover, only two, TROLL-2 and -3, of 8 TROLLs have TAp-3 binding sites. How do you differentially name these two from the others?

Figure 1B. Unlike TROLL2, the expression of TROLL3 is not high in CA1D cell line. Its expression seems to be specific to DCIS cell line. Why did authors choose TROLL3? It seems that TROLL 8 expression parallels TROLL2 and why authors have not considered TROLL8? Moreover, TROLL8 KD had maximum effects on cell migration and invasion! What criteria have been considered for their selection? Their divergent protein coding RNAs do not seem to have common functions: one affects metastasis and the other cancer cell survival.

To check whether TROLL-2/-3 have common functions on global scale, authors should sequence RNA from TROLL-2/-3 knockdown (KD) samples and functionally annotate the differentially expressed protein coding RNAs.

Figure 1C-D: Authors should overexpress TAp63 and check the expression of TROLLs. This further lends support that they are authentic targets of TAp63.

Figure 3A-B and 3e: WDR26 and TROLLs interaction was verified purely based in vitro approaches. Authors should validate WDR26 interaction with TROLL2 and -3 in vivo using ChIRP or other related technologies. Importantly, it would be interesting to know whether WDR26 interacts with a common primary sequence in the two lncRNAs?

Figure 3C-D: Since WDR26 interact with both TROLL2 and 3, and appear to affect common biological functions such as cell migration/invasion, it would be good to knockdown one lncRNA and overexpress the other. I expect that the presence of one lncRNA to the level of both lncRNAs could recapitulate the functions lost as a consequence of knockdown of one of the lncRNAs.

Fig: 1: TROLL-3 KD does not have any effect on proliferation while TROLL2 had minor effect.

However, Supp fig 3J: overexpression of both had effect on cell proliferation. Thus, data from KD and OE contradict each other.

Supp Figure 4e-f: While the presented data (TROLL-3) is interesting, the data is however significant in only one of the two cohorts. Why TROLL-2 data is not presented? To make clinical part of the manuscript more significant, I would like to see detailed clinical significance data for TROLL2 and TROLL3 based on two independent tumor cohorts. Do they independently predict the clinical outcome?

Suppl Figure 6C: The nuclear and cytoplasmic localization of WDR26 following TROLL-2 and -3 is not convincing. Nuclear control HSP90 levels are low in TROLL-2 and -3 samples compared siRNA control. Moreover, two bands are seen in nuclear fraction which are evident in suppl figure 6a. Even though nuclear bands are prominent, only one band is seen.

Figure 6: If TROLL-2/3 are required for promoting interaction between WDR26 and AKT, one would expect AKT as one of the interacting partners in their TROLL interacting partner screening.

Figure 6F: In control cells WDR26 nuclear localization is clear. However, in TROLL-2/3 overexpressing cells WDR26 cytoplasmic localization is not convincing.

ISH was done using Exiqon double digoxigenin labelled LNA probes. A single LNA probe was used for ISH. Is single LNA probe for each lncRNA is sensitive enough to detect lncRNA on tissue microarrays? I cannot find RNase controls. Very important that the signals is specific each lncRNAs. Have the authors tried single molecule RNA FISH on tissue microarrays?

Minor points:

Figure 1A & S1A: Statistical significance for the differentially expressed lncRNA should be included. It seems p value 0.05 and fold change 1.33 X fold was considered, and I cannot find FDR values. Why 1.3 fold change was considered as a cutoff?

Supp fig 2i J: ISH signals should be highlighted with arrow heads.

Data in Figure 4a-b should be validated using nuclear cytoplasmic fractions.

Lanes 257-258: 378 cancers- reads a bit strangely. It should be something like 378 tumor tissues or specimens spanning—

Crispr/Cas9: What kind of deletions/cleavage were created for Troll-2 and -3. A more detailed information in the methods sections would be required.

Reviewer #2 (Remarks to the Author):

Nature Communications manuscript NCOMMS-19-06126
Comments to authors:

Napoli and colleagues describe a novel axis involving TAp63/lncRNAs and AKT in the control of metastasis of breast cancer. The authors identified 8 non-coding RNAs (TROLLs) regulated by TAp63. They focus the work on TROLL-2 and -3 which appear to alter AKT signalling by sequestering the scaffold protein WDR26 in the cytoplasm. The result of this signalling is an alteration of the migration capability of the cancer cells.

A major strength of this study is the wide use of clinical data to validate and/or verify the hypothesis. While this approach provides a strong relevance to the work, it implies the drawback that often the data are only correlative. For these reasons there are a number of experiments listed below to improve the strength of the conclusions. In addition to this, a general concern is that a bit of confusion emerges between the function of the axis TAp63/TROLLs/WDR26 in primary tumours and in metastasis. Some experiments are performed with orthotopic injection in mammary fat-pad, others are performed by tail-vein-injection. Is this signalling important for one or both these processes? The biology at the basis of tumour growth and metastatic spread is very different.

Major points should be addressed:

1- The study starts from the description of a relationship between TROLLs and TAp63, however after figure 1 TAp63 is lost and the story continue exploring the function of TROLL-2 and -3. It is important to demonstrate what is the contribution of TROLL-2 and -3 to the TAp63 mediated breast cancer phenotype, otherwise the paper appears just disconnected with different sets of independent observations.

2- The authors state that they transcribed TROLL-2 and -3 in vitro for the protein array. More information should be provided in regard of the transcriptional initiation and termination as well as splicing isoforms of these lncRNAs expressed in breast cancer. Did the author verify the isoforms they transcribed are physiologically relevant?

3- The authors validate TROLLs/WDR26 interaction with an in vitro pull-down. An RNA immunoprecipitation (RIP) is required in order to validate the data and assess the interaction in a more physiological environment.

4- Is WDR26 regulating metastatic process in vivo? Is deltaNES-WDR26 affecting TROLLs mediated metastatic phenotype in vivo?

5- Related to point 1 and point 4, is WDR26 correlated to TAp63 mediated metastatic suppression in human samples and mouse models?

Minor Points:

- Tail vein injection is not recapitulating a real metastatic process. Authors should rephrase using the terminology "dissemination" or "colonisation" to be more accurate.
- Molecular Makers should be indicated in the figures.

Reviewer #3 (Remarks to the Author):

The authors have made several quite interesting observations from a series of experiments that originated from their long interest in the tumor suppressor TAp63. They found a number of long non-coding RNAs appear to be negatively regulated by TAp63 with their levels increasing upon loss of TAp63. They used single siRNAs to focus their interest on two lncRNAs – TROLL-2 (RPSAP52) and TROLL-3 (TRAF3IP2-AS1) – and present in vitro evidence that they bind to an adaptor protein WDR26 among others. The two lncRNAs appear to be "dragging" WDR26 from the nucleus, and via a to-be-defined mechanism, to increase LPA-induced AKT phosphorylation. This mechanistic data is accompanied by a great deal of data mining of clinical patient data resulting in correlative findings backing their mechanistic model

The manuscript is well written and includes a large body of generally high quality data. However, the scope of the study is very broad and individual component findings appear somewhat shallow

and unconvincing, despite the large amount of data presented. For example, All the mechanistic data is obtained in a single cell line—for publication it should be repeated in several other lines. Similarly, there is little data how the two lncRNAs bind to WDR26 does this interaction actually occur in cells—the data presented in largely in vitro? What is the stoichiometry of the complex—does a single complex contain all three elements—this would be consistent with the fact that even partial knockdown of a single lncRNA can have such a large effect on the invasion assay and cell localization data. What is the evidence that supports the claimed “PTEN-independent AKT activation”? Is the observed WDR26-regulated AKT phosphorylation only found in the setting where serum-starved cells are stimulated with the GPCR ligand LPA? LPA works via p110beta which is often activated where PTEN expression is lost. There should be data using insulin or EGF that activate other PI3K isoforms. What is the fate of PTEN activity in the manipulated cells?

In addition, the manuscript includes a potential technical error, which should be clarified by the authors before the work is considered further. The authors utilized a doxycycline-inducible CRISPR/Cas9 tool to “knock down” genes of interest including those encoding non-coding RNA TROLL-2 (RPSAP52) and TROLL-3 (TRAF3IP2-AS1). They demonstrate that doxycycline induction led to the appearance of approx. 50% indel of the targeted loci (Fig. S2h), reduced RNA signal in derived tumors (Fig. S2i-j), and importantly, an apparent inhibition of tumorigenesis (Fig. 2e), tumor growth (Fig. 2f), and lung metastasis (Fig. 2g-h).

Unfortunately it remains difficult to understand how small indel generated by a single guide RNA, which in this case targeting transcription start sites of the two lncRNAs, could produce a functional knockdown or knockout of the targeted non-coding RNA genes (Note that established methods to CRISPR out non-coding RNAs is typically based on a pair of guides to produce large-fragment deletion). In addition, regarding the observed indel (Fig. S2h), it is also unclear how indel could be generated in cells introduced with sgTROLL-3, give that the guide sequence is not followed by the NGG PAM motif for *S. pyogenes* Cas9.

(The first 23 nucleotides of human TRAF3IP2-AS1 (NR_034108.1) are as follows cgaaggggc gccggagcac CGA. Note that the first 20 nts were used as the guide for gene editing by the authors – as described in the Method section, and “CGA” does not follow the NGG PAM motif used by Cas9).

Response to Reviewers' Comments:

We would like to thank the reviewers for their constructive comments and suggestions. We have addressed all the points raised by the reviewers and believe that the manuscript is of high significance to the cancer and non-coding RNA fields. We now provide additional evidence of the pan-cancer, pro-metastatic roles of TAp63-regulated lncRNAs, *TROLL-2* and *TROLL-3*. In this resubmission, we unveiled a novel role for lncRNAs in cancer cells and demonstrate that *TROLL-2* and *TROLL-3* promote the cytoplasmic localization of WDR26 by preventing its interaction with NOLC1, a shuttling protein that – in the absence of *TROLL-2* and *TROLL-3* – sequesters WDR26 in the nucleus. We also have additional *in vivo* evidence of the pro-metastatic roles of *TROLL-2* and *TROLL-3*. This was shown using shRNAs against either *TROLL-2* and *TROLL-3* using breast cancer, lung adenocarcinoma and melanoma xenograft mouse models. Our specific point-by-point response to each comment is below in boldface type.

Reviewers' comments:

Reviewer #1:

This study characterizes TAp63 regulated lncRNAs (TROLLs). P63 is a p53 ortholog plays an important role in the suppression of tumor metastasis and invasion. By using locus conservation approach, authors have chosen 8 lncRNAs (TROLLs) out of 54 differentially expressed lncRNAs. For further functional investigations, 2 lncRNAs, TROLL-2/-3, have been chosen based on consensus TAp63 bindings sites in their promoter regions and ChIP-qPCR data. TROLLs expression correlates breast cancer progression, and their higher expression correlates with cytoplasmic localization of their common interacting protein WDR26 and activation of pro-survival pathways PI3K/AKT pathways. Overall, this is an interesting study provide insights into lncRNA dependent gene regulatory mechanisms underlying the tumor progression. I have several concerns listed below.

Major comments:

1. Figure 1A and 1E: Recently, MALAT1 has recently been shown to suppress breast cancer by attenuating the function of metastatic promoting factor TEAD (Kim J et al., 2018 Dec. Nature

Genetics). However, in the current study *TAp63* KD results in the activation of *MALAT1*, and increased cell migration/invasion. How do these two contrasting observations on *MALAT1* reconcile?

Malat1 is an important lncRNA with context dependent functions. Specifically, murine Malat1 has been shown to both promote (Arun et al., *Genes Dev.* 2016 Jan 1;30(1):34-51) and suppress (Kim et al., *Nat Genet.* 2018 Dec;50(12):1705-1715) breast cancer metastasis, and the respective authors hypothesised that this variability may be ascribed to the different knockout mouse models and mouse backgrounds used. In the case of human Malat1, however, there is more consensus on its function as an oncogene in breast cancers. Indeed, there are multiple reports from different labs showing that in human breast cancer patients high Malat1 levels correlate with higher risk of relapse and reduced overall survival (Huang et al., *Oncotarget.* 2016 Jun 21;7(25):37957-37965; Wang et al., *Breast Cancer Res Treat.* 2018 Sep;171(2):261-271; and Zheng et al., *Biosci Rep.* 2019 Feb 15;39(2). pii: BSR20181284. doi: 10.1042/BSR20181284). In line with these observations, our data show that Malat1 levels are higher in human breast cancer cells compared to normal cells (see Fig. 1b and Supplementary Fig. 1a), which is a trend observed also in our mouse model (see Fig. 1a and 1c). This point has been added to the manuscript on page 5.

2. Most of these lncRNA already have names and why they should be renamed with TROLLs. It only confuses the research community with several aliases for a single lncRNA. Moreover, only two, TROLL-2 and -3, of 8 TROLLs have TAp-3 binding sites. How do you differentially name these two from the others?

Among the identified lncRNAs, Malat1 is the only one whose name is the same for both the mouse and the human orthologues. In the case of the remaining lncRNAs, the two species have different names, with most of the mouse names being the sequence ID from the Riken mouse genome project (e.g. 1700006J14Rik, E130307A14Rik, 2610307P16Rik, and 2410006H16Rik). To emphasize the fact that the identified lncRNAs are conserved between mouse and human and that their expression is regulated by TAp63, we renamed them as

TAp63 regulated oncogenic lncRNAs (or TROLLs). However, the original name of their sequence is shown (see Fig. 1a,b).

To clarify the fact that all the identified lncRNAs are *bona fide* TAp63 direct target genes, we have analysed their respective promoters by identifying putative TAp63 binding sites (Table 1 and also included as new Supplementary Fig. 1h).

Element	Location	TAp63 Binding Site	MM / spacer
TROLL-1 promoter	-2213 to -2238	acaCATGctg tgtgcc aggCATGccc	2 / 6
TROLL-2 promoter	-4269 to -4240	aggCATGttc tagaaagaa agtCATGtac	2 / 9
TROLL-3 promoter	-3825 to -3800	aggCATGagc cacagc accCATGccc	4 / 6
TROLL-4 promoter	-4898 to -4919	gggCATGtca a aatCATGcac	3 / 1
TROLL-5 promoter	-2297 to -2320	ggaCATGcgc ctg tgcCATGccc	3 / 3
TROLL-6 promoter	-2264 to -2287	cggCATGacg tgt tttCATGact	6 / 3
TROLL-7 promoter	-3851 to -3874	cttCATGtct gac ctgCATGtct	5 / 3
TROLL-8 promoter	-2013 to -2034	gaaCATGatc g atcCATGtca	3 / 1
TROLL-9 promoter	-3254 to -3278	gggCATGgcc ttaa gcaCATGggc	4 / 4

Table 1. Table listing the TAp63 binding sites on the promoters of the 9 lncRNAs. Number of mismatches to the TAp63 consensus binding site are shown in red. MM indicates the number of mismatches to the TAp63 consensus binding site. Spacer indicates the number of nucleotides between two half sites.

These sites have been validated by CHIP-qRT-PCRs, confirming that TAp63 is recruited to the promoters of all 9 lncRNAs (Figure 1 and also included as new Supplementary Fig. 1i).

Figure 1. qRT-PCR of TAp63 ChIP assays on the promoters of the indicated lncRNAs.

Additionally, down-regulation of TAp63 (Fig. 1c,d) and up-regulation of TAp63 (Figure 2 and also included as new Supplementary Fig. 1d-g) have opposite effects on the expression of the lncRNAs in both mouse and human cell lines, thus indicating that all 9 lncRNAs are novel *bona fide* direct target genes of TAp63.

Figure 2. WT MECs and CA1D cells were transfected with Myc-TAp63 γ or empty vector as a negative control. 48hr after transfection, the cells were then tested by WB analysis to confirm TAp63 overexpression (left) and by qRT-PCR to check for the levels of the indicated lncRNAs. Data are mean \pm SD, n = 3, * versus empty vector, $P < 0.05$, two-tailed t-test.

3. Figure 1B. Unlike TROLL2, the expression of TROLL3 is not high in CA1D cell line. Its expression seems to be specific to DCIS cell line. Why did authors choose TROLL3? It seems that TROLL 8 expression parallels TROLL2 and why authors have not considered TROLL8? Moreover, TROLL8 KD had maximum effects on cell migration and invasion! What criteria have been

considered for their selection? Their divergent protein coding RNAs do not seem to have common functions: one affects metastasis and the other cancer cell survival.

We pursued *TROLL-2* and *TROLL-3* based on two criteria:

- 1) uniqueness → among the identified lncRNAs, they are the only 2 to be antisense, while the remaining lncRNAs are intergenic;**
- 2) guilt-by-association → their respective antisense protein coding genes (HMGA2 for *TROLL-2* and TRAF3IP2, also known as ACT1, for *TROLL-3*) are both known oncogenes supporting tumour growth and dissemination (e.g. Sgarra et al., *Biochim Biophys Acta Rev Cancer*. 2018 Apr;1869(2):216-229 for HMGA2; and Alt et al., *Oncotarget*. 2018 Jul 3;9(51):29772-29788 for TRAF3IP2).**

The manuscript has now been modified to clarify both criteria of selection and is included on page 7.

*4. To check whether *TROLL-2/-3* have common functions on global scale, authors should sequence RNA from *TROLL-2/-3* knockdown (KD) samples and functionally annotate the differentially expressed protein coding RNAs.*

We thank the reviewer for the suggestion. However, we chose to identify lncRNA-protein interactions via protein micro-array experiments (see Fig. 3a,b), because lncRNAs are known to exert their functions by interacting with specific proteins that subsequently act as the ultimate effectors of the lncRNA's functions (e.g. RNA binding proteins, chromatin remodelling factors, *et cetera*) (Salehi et al., *J Cell Mol Med*. 2017;21:3120-3140 and Rao et al., *Mol Biol Rep*. 2017;44:203-218). Therefore, the protein micro-array approach gave us the opportunity to obtain insight on the common mechanism of function of *TROLL-2* and *TROLL-3* and to identify WDR26 as their common downstream effector, ultimately providing a therapeutic opportunity for p53 mutant cancers.

5. Figure 1C-D: Authors should overexpress TAp63 and check the expression of TROLLs. This further lends support that they are authentic targets of TAp63.

As suggested by the reviewer, we tested the effect of the overexpression of TAp63 on the expression levels of the lncRNAs as described above (please see data in Figure 2 and new Supplementary Fig. 1d-g).

6. Figure 3A-B and 3e: WDR26 and TROLLs interaction was verified purely based in vitro approaches. Authors should validate WDR26 interaction with TROLL2 and -3 in vivo using ChIRP or other related technologies. Importantly, it would be interesting to know whether WDR26 interacts with a common primary sequence in the two lncRNAs?

To address the reviewer's comment, we performed a CLIP experiment by immunoprecipitating endogenous WDR26 from the CA1D cells and assessed for the enrichment of TROLL-2 and TROLL-3 in the eluate. As shown in figure 3 (included in the manuscript as new Supplementary Fig. 6v'), endogenous WDR26 interacts with both lncRNAs.

Figure 3. qRT-PCR for TROLL-2 and TROLL-3 in eluates of CLIP-ed WDR26 in CA1D cells transfected with the indicated siRNAs. Data are mean \pm SD, n = 3, * versus siControl, $P < 0.05$, two-tailed t-test.

Additionally, to identify the sequence in the two lncRNAs necessary for their interaction with WDR26, we performed the CLIP experiment in the presence of RNase, so that only the regions interacting with WDR26 (and therefore protected by the RNase treatment) could then be detected by qRT-PCR. This approach allowed us to map the regions involved in the interaction (see Figure 4 also included in the manuscript as new Supplementary Fig. 6x',y'). The region of *TROLL-2* protected by the RNase treatment is within nucleotide 403 and 627, while that of *TROLL-3* is within nucleotide 360 and 603.

Figure 4. qRT-PCR for overlapping fragments of *TROLL-2* and *TROLL-3* in RNase-treated eluates of CLIP-ed WDR26 in CA1D cells. Data are mean \pm SD, n = 3.

Looking for a possible similarity between these two regions, we compared them using Blast (<https://blast.ncbi.nlm.nih.gov/Blast.cgi>). Notably, we found that they contain a common sequence (corresponding to nucleotides 522-538 on *TROLL-2* and 467-482 on *TROLL-3*). To assess if this common sequence is mediating the interaction between the lncRNAs and WDR26, we generated deletion mutants (*TROLL-2* Δ 522-538 and *TROLL-3* Δ 467-482) and used them in pull-down assays to assess their ability to interact with endogenous WDR26 in CA1D cells. As shown in Figure 5 (also included in the manuscript as new Supplementary Fig. 6z',a''), deletion of these regions significantly impairs the interaction of these lncRNAs with WDR26. Altogether, these data indicate that endogenous WDR26 binds to *TROLL-2*

and *TROLL-3* in CA1D cells and that this interaction is mediated by a common sequence present in both lncRNAs.

Figure 5. Representative WB analysis for WDR26 in pull-down assays performed in CA1D cells using the in vitro transcribed and biotinylated sense and antisense strands of the indicated lncRNAs. 3 biological replicates were performed and the interaction was quantified relatively to the respective input. Data are mean \pm SD, n = 3, * versus sense strand of the full-length lncRNA, $P < 0.05$, two-tailed t-test.

7. *Figure 3C-D: Since WDR26 interact with both TROLL2 and 3, and appear to affect common biological functions such as cell migration/invasion, it would be good to knockdown one lncRNA and overexpress the other. I expect that the presence of one lncRNA to the level of both lncRNAs could recapitulate the functions lost as a consequence of knockdown of one of the lncRNAs.*

Following the reviewer's suggestion, we performed cell migration and invasion assays in CA1D cells transfected with the siRNAs for one lncRNA and the plasmid to overexpress the other lncRNA. As shown in Figure 6 (also included in the manuscript as Supplementary Fig.

6d'',e''), the overexpression of one lncRNA does not rescue the absence of the other one. This is likely due to the fact that these lncRNAs form a trimeric complex with WDR26 (see below our response to reviewer #3's comment on the stoichiometry of the complex, Figure 16 also included in the manuscript as Supplementary Fig. 6b'',c''). Given that lack of one lncRNA impairs the interaction between WDR26 and the remaining lncRNA, both lncRNAs are necessary to promote the pro-oncogenic activities downstream of WDR26, including cell migration and invasion.

Figure 6. Cell migration (left) and invasion (right) assays performed in CA1D cells transfected with the indicated constructs. Data are mean \pm SD, n = 3, * versus siControl, # versus the respective empty, $P < 0.05$, two-tailed t-test.

8. *Fig:1: TROLL-3 KD does not have any effect on proliferation while TROLL2 had minor effect. However, Supp fig 3J: overexpression of both had effect on cell proliferation. Thus, data from KD and OE contradict each other.*

We think that the apparent contradiction is due to the normalization used in the 2 experiments. If we normalized each graph to the respective control, so that both siControl in Fig. 1g and Supplementary Fig. 3J have a proliferation equal to 100%, the other samples would be as follows:

_ siTROLL-2 has a reduction of 32%, while the overexpression of TROLL-2 has an increase of 24%;

_ siTROLL-3 has a reduction of 13% (not statistically significant), while O/E of TROLL-2 has an increase of 17% (statistically significant).

9. *Supp Figure 4e-f: While the presented data (TROLL-3) is interesting, the data is however significant in only one of the two cohorts.*

The data shown in Supplementary Fig. 4e,f (TCGA basal-like breast cancer cohort, now Supplementary Fig. 4g,h) and Supplementary Fig. 5a',b' (TCGA melanoma cohort, now Supplementary Fig. 5a',b') show that WDR26 levels stratify patients in groups with better or worse survival only in tumours with high levels of *TROLL-3* (left panels) but not in tumours with low levels of *TROLL-3* (right panels). These data suggest that only when *TROLL-3* levels are high, and therefore WDR26 localizes in the cytoplasm as shown in multiple TMAs (see Fig. 5a), tumours are more aggressive and associated to reduced patient survival. We have now edited the manuscript and clarified the description of these findings on page 12.

10. *Why TROLL-2 data is not presented?*

When we assessed the same TCGA cohorts based on the levels of *TROLL-2*, the data did not reach statistical significance. We have now added a comment to the manuscript related to this on page 12.

11. To make clinical part of the manuscript more significant, I would like to see detailed clinical significance data for TROLL2 and TROLL3 based on two independent tumor cohorts. Do they independently predict the clinical outcome?

Following the reviewer’s suggestion, we assessed the prognostic value of both lncRNAs in a cohort of breast cancer patients and melanoma patients (matching the “Moffitt breast cancer TMA” and “Moffitt melanoma TMA” data shown in Fig. 5a). As shown in Figure 7, also included in the manuscript as Fig. 2e,f (breast cancer) and Supplementary Fig. 5w,x (melanoma), high levels of either lncRNA correlates with reduced overall survival in both cohorts, thus indicating that both lncRNAs are prognostic factors in these two tumour types.

Figure 7. Kaplan-Meier curves of overall survival in breast cancer patients (top panels) and melanoma patients (bottom panels) showing the prognostic value of the indicated lncRNA in tumours with higher or lower than median levels.

12. *Suppl Figure 6C: The nuclear and cytoplasmic localization of WDR26 following TROLL-2 and -3 is not convincing. Nuclear control HSP90 levels are low in TROLL-2 and -3 samples compared siRNA control. Moreover, two bands are seen in nuclear fraction which are evident in suppl figure 6a. Even though nuclear bands are prominent, only one band is seen.*

We have now found that the doublet observed in some western blots for WDR26 is due to a non-specific band detected by some of old lots of the WDR26 antibody (Abcam, ab85961). Therefore, we have now repeated all the cell fractionation experiments with a new lot from Abcam. As shown in Figure 8 (also included in the manuscript as new Fig.6a and Supplementary Fig. 6c), the new lot of the WDR26 antibody recognises only one band. To take into account any difference in the loading among the samples, the WDR26 bands are normalized based on the H3 bands (nuclear fractions) and the HSP90 bands (cytoplasmic fractions). These data show a clear shift to nuclear localization after *TROLL-2* and *TROLL-3* knock down.

Figure 8. Representative western blot (left) and quantification of the percentage (right) of WDR26 localization in the nuclei (Nuc) and cytosols (Cyt) of the indicated cell lines. Histone H3 and HSP90 were used as controls, respectively. Data are mean \pm SD and analysed with two-way ANOVA. n = 3, * vs. siControl, P < 0.005.

13. Figure 6: If *TROLL-2/3* are required for promoting interaction between *WDR26* and *AKT*, one would expect *AKT* as one of the interacting partners in their *TROLL* interacting partner screening.

Although *AKT* is one of the proteins present in the Protoarray Human Protein Microarrays v5.0 (ThermoFisher Scientific), the condition of this assay may not be optimal for the interaction between the lncRNAs and *AKT*, given also the fact that *AKT* does not have either a canonical or a non-canonical RNA binding domain. To overcome this caveat and assess the interaction of *TROLL-2* and *TROLL-3* with *AKT* in a more physiological context, we performed a CLIP assay in CA1D cells. As shown in Figure 9, also included as Supplementary Fig. 6w', endogenous *AKT* binds to the lncRNAs in a *WDR26* dependent manner. This data indicates that *WDR26* is required to mediate the interaction between *AKT* and the two lncRNAs, thus explaining why in the protein microarray assay – where only one recombinant protein is present in each spot – the interaction was not detected.

Figure 9. qRT-PCR for *TROLL-2* and *TROLL-3* in eluates of CLIP-ed *AKT* in CA1D cells transfected with the indicated siRNAs. Data are mean \pm SD, n = 3, * versus siControl, $P < 0.05$, two-tailed t-test.

14. *Figure 6F: In control cells WDR26 nuclear localization is clear. However, in TROLL-2/3 overexpressing cells WDR26 cytoplasmic localization is not convincing.*

The panel has now been replaced and appears in the manuscript as new Fig. 6h.

14. ISH was done using Exiqon double digoxigenin labelled LNA probes. A single LNA probe was used for ISH. Is single LNA probe for each lncRNA is sensitive enough to detect lncRNA on tissue microarrays? I cannot find RNase controls. Very important that the signals is specific each lncRNAs. Have the authors tried single molecule RNA FISH on tissue microarrays?

The probes used in the ISH assays are labelled with digoxigenin (DIG) both at the 5' and 3' ends. Compared to FAM-labelled probes, whose signal can be below the detection threshold unless the lncRNA of interest is expressed at sufficiently high levels, the presence of DIG allows for the detection of the probe by an anti-digoxigenin antibody conjugated with alkaline phosphatase (AP), whose chromogenic reaction amplifies the signal thus increasing the sensitivity of the assay.

To confirm the specificity of our probes, we have now added the controls with tumours expressing the shRNA for the lncRNA of interest as well as the ISH using a scramble probe as negative control (Figures 10-12, also included as new Supplementary Fig. 2h,i and Supplementary Fig. 5c'-f').

Figure 10. Representative images of ISH for *TROLL-2* and *TROLL-3* in breast tumours derived from CA1D (top panels) and MDA MB-231 cells (bottom panels) infected with the indicated shRNAs. The LNA probe was detected and visualized via a chromogenic reaction (purple), while Nuclear Fast Red™ was used as a counterstain. A scramble LNA probe was used as a negative control.

Figure 11. Representative images of ISH for *TROLL-2* and *TROLL-3* in lung adenocarcinomas derived from H1299 (top panels) and H358 cells (bottom panels) infected with the indicated shRNAs. The LNA probe was detected and visualized via a chromogenic reaction (purple), while Nuclear Fast Red™ was used as a counterstain. A scramble LNA probe was used as a negative control.

Figure 12. Representative images of ISH for *TROLL-2* and *TROLL-3* in melanomas derived from A375 (top panels) and Malme-3M cells (bottom panels) infected with the indicated shRNAs. The LNA probe was detected and visualized via a chromogenic reaction (purple), while Nuclear Fast Red™ was used as a counterstain. A scramble LNA probe was used as a negative control.

Minor points:

15. Figure 1A & S1A: Statistical significance for the differentially expressed lncRNA should be included. It seems p value 0.05 and fold change 1.33 X fold was considered, and I cannot find FDR values. Why 1.3 fold change was considered as a cutoff?

As suggested by the reviewer, we have now utilised a more stringent fold change (1.5x) and included the FDR values in our analysis of the RNA-sequencing data. As now described in the Methods, we first used the R package limma to identify protein coding and non-coding RNAs that were differentially expressed in the RNA-sequencing data of the MCF10A breast cancer progression model (MCF10A vs. AT1, AT1 vs. DCIS, and DCIS vs. CA1D) using the fold change cut-off of 1.5x and the FDR-adjusted p-value of 0.1. By selecting pairs of coding/non-coding RNAs within 100kb from each other, we obtained 882 human coding genes, and 540 non-coding RNAs. We next considered the corresponding 890 mouse coding genes, and their neighbouring 591 noncoding RNAs. We used limma to analyse RNAs differently expressed between WT and *TAp63*^{-/-} MECs. Using the cut-off p-value<0.05 and fold change exceeding 1.5x, and the genomic distance of at most 100kb from a coding gene and neighbouring non-coding RNA, we obtained 11 coding genes and 12 noncoding RNAs. The mouse non-coding RNAs were further validated via RT-qPCR, and those that passed, as well as their human counterparts, were depicted graphically as heatmaps using the Python SciPy scientific library (see Figure 13, also included as Fig. 1a,b).

Figure 13. Heatmap visualization of the 9 conserved lncRNAs differentially expressed in WT and *TAp63*^{-/-} mammary epithelial cells (left) and in the MCF10A breast cancer progression model (right).

16. *Supp fig 2i J: ISH signals should be highlighted with arrow heads.*

The figure has been modified as request.

17. *Data in Figure 4a-b should be validated using nuclear cytoplasmic fractions.*

We have now clarified in the text that Fig. 4a,b are representative images of IHC for WDR26 performed in a TMA of breast cancer progression (US Biomax, BR480a). Given the nature of these samples (i.e. small core of biopsies), it is challenging to perform cell fractionation from them. However, the change in localization of WDR26 across breast cancer progression observed in this TMA is corroborated by the cell fractionation data obtained with the MCF10A breast cancer progression model, included in the manuscript as new Fig.6a and Supplementary Fig. 6a.

18. *Lanes 257-258: 378 cancers- reads a bit strangely. It should be something like 378 tumor tissues or specimens spanning???*

The text has been replaced with “378 tumour specimens, comprising 51 ovarian...”. This text is on page 13.

19. *Crispr/Cas9: What kind of deletions/cleavage were created for Troll-2 and -3. A more detailed information in the methods sections would be required.*

Based on the suggestion provided by reviewer #3 (see below), the data with the CRISPR/Cas9 system has been replaced with the experiments using shRNA against the lncRNA of interest.

Reviewer #2:

Nature Communications manuscript NCOMMS-19-06126

Comments to authors:

Napoli and colleagues describe a novel axis involving TAp63/lncRNAs and AKT in the control of metastasis of breast cancer. The authors identified 8 non-coding RNAs (TROLLs) regulated by TAp63. They focus the work on TROLL-2 and -3 which appear to alter AKT signalling by sequestering the scaffold protein WDR26 in the cytoplasm. The result of this signalling is an alteration of the migration capability of the cancer cells. A major strength of this study is the wide use of clinical data to validate and/or verify the hypothesis. While this approach provides a strong relevance to the work, it implies the drawback that often the data are only correlative. For these reasons there are a number of experiments listed below to improve the strength of the conclusions.

1. In addition to this, a general concern is that a bit of confusion emerges between the function of the axis TAp63/TROLLs/WDR26 in primary tumours and in metastasis. Some experiments are performed with orthotopic injection in mammary fat-pad, others are performed by tail-vein-injection. Is this signalling important for one or both these processes? The biology at the basis of tumour growth and metastatic spread is very different.

We would like to thank the reviewer for acknowledging the strength added to our manuscript by incorporating clinical data (723 biopsies from 10 different TMAs and corresponding to 6 different cancer types). It is indeed the analysis of these TMAs that clearly indicated that levels of *TROLL-2* and *TROLL-3* increase along the progression of these 6 cancers (see Fig. 5a) and that *TROLL-2* and *TROLL-3* are relevant for both tumour and metastasis formation. Given the correlative nature of these data, we have now tested the *in vivo* role of these lncRNAs in two different xenograft models of breast cancer (CA1D and MDA MB-231 cells) as well as two different models of lung adenocarcinoma (H1299 and H358 cells) and melanoma (A375 and Malme-3M). As shown in Figures 14-16, also included in the manuscript as Fig. 2g-n and Fig. 5h-w, knockdown of either *TROLL-2* or *TROLL-3* via shRNA reduces the tumour formation and the lung colonization (metastatic progression) ability of these cancers, thus supporting the importance of *TROLL-2* and *TROLL-3* in both tumour growth and metastatic spread.

Figure 14. Representative H&E images of breast tumours (top panels) and lung colonies (bottom panels) derived from the indicated cells. Quantification of tumour volumes and of the percentage of lung colonization is shown in the right panels. n = 10 (breast tumours) and 5 (lungs). * vs. shNT, $P < 0.005$, two-tailed Student's t test.

Figure 15. Representative H&E images of lung adenocarcinomas (top panels) and lung colonies (bottom panels) derived from the indicated cells. Quantification of tumour volumes and of the percentage of lung colonization is shown in the right panels. $n = 5$ lungs for all groups. * vs. shNT, $P < 0.005$, two-tailed Student's t test.

Figure 16. Representative H&E images of melanomas (top panels) and lung colonies (bottom panels) derived from the indicated cells. Quantification of tumour volumes and of the percentage of lung colonization is shown in the right panels. $n = 10$ (breast tumours) and 5 (lungs). * vs. shNT, $P < 0.005$, two-tailed Student's t test.

2. *The study starts from the description of a relationship between TROLLs and TAp63, however after figure 1 TAp63 is lost and the story continue exploring the function of TROLL-2 and -3. It is important to demonstrate what is the contribution of TROLL-2 and -3 to the TAp63 mediated breast cancer phenotype, otherwise the paper appears just disconnected with different sets of independent observations.*

To demonstrate the pro-oncogenic role of the two lncRNAs in a TAp63 mediated breast cancer phenotype, we have now taken advantage of the MDA MB-231 cell line, an aggressive model of breast cancer whose growth in vivo relies on the inhibition of the tumour and metastasis suppressor TAp63 by mutant p53 (Adorno et al., Cell. 2009 Apr 3;137(1):87-98; and Muller et al., Cell. 2009 Dec 24;139(7):1327-41). As shown above in Figure 14, also included in the manuscript as Fig. 2k-n, reduction in the levels of *TROLL-2* and *TROLL-3* via shRNA significantly decreases the ability of these cells to form tumours in vivo and to colonize the lungs. These data, therefore, suggest that the breast cancer phenotype observed in a context where TAp63 is inhibited is mediated, at least in part, by *TROLL-2* and *TROLL-3*.

3. *The authors state that they transcribed TROLL-2 and -3 in vitro for the protein array. More information should be provided in regard of the transcriptional initiation and termination as well as splicing isoforms of these lncRNAs expressed in breast cancer. Did the author verify the isoforms they transcribed are physiologically relevant?*

The NCBI database reports only 1 isoform for *TROLL-2*, also known as RPSAP52, whose ID is NR_026825.2 (<https://www.ncbi.nlm.nih.gov/gene/?term=RPSAP52>). Regarding *TROLL-3*, also known as TRAF3IP2-AS1, 4 isoforms are reported (<https://www.ncbi.nlm.nih.gov/gene/643749>). The longest isoform, NR_034108.1, is the one identified as differentially expressed in the MCF10A breast cancer progression model and was used for the overexpression experiments (Fig. 3c,d and Supplementary Fig. 3a-k, Fig. 6b and 6d, and Supplementary Fig. 6e-i and 6d'',e''), for the in vitro transcription in the protein microarray experiment (Fig. 3a,b) and in the pull-down assays (Fig. 3e and Supplementary Fig. 6z'-c'). The text has now been modified to include this information on page 10.

4. *The authors validate TROLLs/WDR26 interaction with an in vitro pull-down. An RNA immunoprecipitation (RIP) is required in order to validate the data and assess the interaction in a more physiological environment.*

As indicated in our response to reviewer #1 (see Figure 3 above), we have now performed a CLIP assay to confirm that endogenous WDR26 and the two lncRNAs interact in CA1D cells. This figure has been added to the manuscript as new Supplementary Fig. 6v'.

5. *Is WDR26 regulating metastatic process in vivo? Is deltaNES-WDR26 affecting TROLLs mediated metastatic phenotype in vivo?*

While we agree that the export of WDR26 to the cytoplasm is crucial for the metastatic process, as indicated both by our *in vitro* data (see Fig. 6d and Supplementary Fig. 6g) and by the positive correlation between the cytoplasmic localization of WDR26 and cancer progression observed in 6 different tumour types (see Fig. 5a), the focus of this manuscript is to understand the functions of *TROLL-2* and *TROLL-3* and to characterize their interaction with WDR26 for the potential of designing future therapies that block the shuttling of WDR26 to the cytoplasm. In line with this goal, we have now clearly demonstrated that *TROLL-2* and *TROLL-3* promote the cytoplasmic localization of WDR26 by preventing its interaction with the shuttling protein NOLC1 and the subsequent nuclear sequestration. These new data are herein shown as Figures 19 and 20 (also included in the manuscript as Fig. 6b,c) and support our proposed model shown in Figure 21 (also included in the manuscript as Fig. 6j).

6. *Related to point 1 and point 4, is WDR26 correlated to TAp63 mediated metastatic suppression in human samples and mouse models?*

To assess the connection between TAp63 and WDR26 in human samples, we performed IHC analysis for TAp63 in a TMA of breast cancer progression (US Biomax, BR480a). In

agreement with previous research by our laboratory in another breast cancer TMA (Su et al., Nature 2010 Oct 21;467(7318):986-90), the quantification of the IHC score of TAp63 by the Oncotopix® software (Visiopharm) shows that the levels of TAp63 are lower in breast cancers compared to normal breast tissue. Additionally, WDR26 levels and cytoplasmic localization inversely correlate with TAp63 levels in this breast cancer TMA. These new data, shown below as Figure 17, have been included in the manuscript as new Supplementary Fig. 4e,f.

Figure 17. Quantification of the IHC score of TAp63 (left) and correlation between the IHC scores of TAp63 and WDR26 (right) in a tissue microarray of breast cancer progression (US Biomax, BR480a). Data were analysed with two-way ANOVA. * vs. normal breast tissue (NB), $P < 0.005$. § vs. lobular hyperplasia (LH), $P < 0.005$. [Please, note that a logarithmic scale is used in the x axis of the right graph]

Minor Points:

7. Tail vein injection is not recapitulating a real metastatic process. Authors should rephrase using the terminology ???dissemination??? or ???colonisation??? to be more accurate.

As suggested by the reviewer, we have now edited the text and described the data as indication of “lung colonization” throughout the manuscript.

8. Molecular Makers should be indicated in the figures.

As suggested by the reviewer, molecular markers were added to the western blotting panels.

Reviewer #3:

The authors have made several quite interesting observations from a series of experiments that originated from their long interest in the tumor suppressor TAp63. They found a number of long non-coding RNAs appear to be negatively regulated by TAp63 with their levels increasing upon loss of TAp63. They used single siRNAs to focus their interest on two lncRNAs ??? TROLL-2 (RPSAP52) and TROLL-3 (TRAF3IP2-AS1) ??? and present in vitro evidence that they bind to an adaptor protein WDR26 among others. The two lncRNAs appear to be ???dragging??? WDR26 from the nucleus, and via a to-be-defined mechanism, to increase LPA-induced AKT phosphorylation. This mechanistic data is accompanied by a great deal of data mining of clinical patient data resulting in correlative findings backing their mechanistic model.

The manuscript is well written and includes a large body of generally high quality data. However, the scope of the study is very broad and individual component findings appear somewhat shallow and unconvincing, despite the large amount of data presented. For example, All the mechanistic data is obtained in a single cell line???for publication it should be repeated in several other lines. Similarly, there is little data how the two lncRNAs bind to WDR26 does this interaction actually occur in cells???the data presented in largely in vitro? What is the stoichiometry of the complex???does a single complex contain all three elements???this would be consistent with the fact that even partial knockdown of a single lncRNA can have such a large effect on the invasion assay and cell localization data.

1. We would like to thank the reviewer for considering our data quite interesting and of high quality. As detailed in our response to reviewer #1 (see Figure 3, also included in the manuscript as new Supplementary Fig. 6v'), we have now performed a CLIP assay in CA1D cells demonstrating that endogenous WDR26 and the two lncRNAs interact in vivo. Additionally, we have now mapped the regions of the lncRNAs binding to WDR26 (see Figure 4 also included as new Supplementary Fig. 6x',y'). Notably, these regions contain a common string of nucleotides (522-538 in TROLL-2 and 467-482 in TROLL-3), whose deletion impairs the interaction between the lncRNAs and WDR26 (see Figure 5, also included in the manuscript as new Supplementary Fig. 6z',a").

2. To determine whether WDR26 and the two lncRNAs are forming a single complex, we have performed a pull-down assay in CA1D cells transfected with siRNA against either lncRNA. As shown in Figure 18, also included in the manuscript as new Supplementary Fig. 6b'',c'', reduced levels of either lncRNA impaired the interaction between WDR26 and the remaining lncRNA, thus suggesting that both lncRNAs interact at the same time with WDR26 possibly forming a trimeric complex. As a biological consequence, the overexpression of one lncRNA cannot rescue the decrease in cell migration and invasion due to the down-regulation of the other lncRNA (see answer to reviewer #1 describing Figure 6, also included in the manuscript as Supplementary Fig. 6d'',e'').

Figure 18. Representative WB analysis for WDR26 in pull-down assays performed in CA1D cells transfected with the indicated siRNAs using the in vitro transcribed and biotinylated sense and antisense strands of the indicated lncRNAs. 3 biological replicates were performed and the interaction was quantified relatively to the respective input. Data are mean \pm SD, n = 3, * versus siControl of the respective sense strand, $P < 0.05$, two-tailed t-test.

3. To better define the mechanism controlling WDR26 localization, we have performed a mass spectrometry analysis comparing proteins that interact with cytoplasmic WDR26 (i.e. WDR26 Δ NLS) vs. nuclear WDR26 (i.e. WDR26 Δ NES). Intriguingly, the top-ranking protein interacting exclusively with WDR26 Δ NES was NOLC1 (also known as Nopp140), a protein known to shuttle between nucleus and cytoplasm (Meier et al., *Cell*. 1992;70:127-138) and to affect the localization of several proteins (Chen et al., *Mol Cell Biol*. 1992;19(12):8536-8546; Li et al., *J Biol Chem*. 1997;272(6)3773-3779; Yuan et al., *Cell Death Discov*. 2017;3:17043). Therefore, we tested whether WDR26 and NOLC1 interact in vivo and if this interaction may be affected by the lncRNAs. As shown in Figure 19, also included in the manuscript as Fig. 6b, NOLC1 interacts with WDR26 more efficiently in MCF10A cells, where WDR26 is mainly nuclear, than in CA1D cells, where WDR26 is mainly cytoplasmic (see Supplementary Fig. 6a). Furthermore, this interaction is affected by *TROLL-2* and *TROLL-3*. Indeed, the overexpression of these lncRNAs in MCF10A cells counteracts the interaction between NOLC1 and WDR26. On the other hand, silencing these lncRNAs in CA1D cells promotes the interaction between these two proteins.

Figure 19. Representative WB analysis of the co-immunoprecipitation using the indicated antibodies in MCF10A cells transfected with both lncRNAs or the empty vector as a control (left) and in CA1D cells transfected either with siRNAs against both lncRNAs or with siControl (right).

Since NOLC1 affects the localization of multiple proteins (Chen et al., *Mol Cell Biol*. 1992;19(12):8536-8546; Li et al., *J Biol Chem*. 1997;272(6)3773-3779; Yuan et al., *Cell Death Discov*. 2017;3:17043), we asked whether the interaction observed in Figure 18 is important

for the localization of WDR26. We silenced NOLC1 in MCF10A cells, where WDR26 efficiently interacts with NOLC1 and is mainly nuclear, and analysed the levels of WDR26 in the nuclear and cytoplasmic fractions. As shown in Figure 20, also included in the manuscript as Fig. 6c, down-regulation of NOLC1 causes a change in the localization of WDR26, which becomes predominantly cytoplasmic.

Figure 20. Representative western blot analysis for WDR26 in the nuclear (Nuc) and cytoplasmic (Cyt) fractions of MCF10A cells transfected with the indicated siRNAs. Histone H3 and HSP90 were used as controls, respectively.

Based on all our data, we now propose the model depicted in Figure 21, also included in the manuscript as Fig. 6j.

Figure 21. In normal cells (e.g. MCF10A cells) the tumour and metastasis suppressor TAp63 inhibits the expression of *TROLL-2* and *TROLL-3*, while NOLC1 interacts with WDR26 and promotes the accumulation of WDR26 in the nucleus. In cancer cells (e.g. CA1D cells), instead, mutant p53 inhibits TAp63, thus allowing for

the expression of *TROLL-2* and *TROLL-3*. These lncRNAs counteract the interaction between NOLC1 and WDR26, while promoting the binding of WDR26 to AKT. As a consequence, the PI3K/AKT pathway is activated and can sustain tumour formation and progression.

4. To extend our mechanistic data regarding the role of *TROLL-2* and *TROLL-3* beyond what we found in CA1D cells, we have now performed experiments in 5 additional xenograft models: another breast cancer model (MDA MB-231 cells), two models of lung adenocarcinoma (H1299 and H358 cells) and two of melanoma (A375 and Malme-3M). In line with our analysis of 723 biopsies from 6 different tumour types showing that high levels of *TROLL-2* and *TROLL-3* correlate with cancer progression (see Fig. 5a), our new in vivo data demonstrate that knockdown of *TROLL-2* and *TROLL-3* strongly decreases the tumour growth and the lung colonization in all the cancer models tested. These new data are shown as Figures 14-16 and are also included in the manuscript as Fig. 2g-n and Fig. 5h-w.

5. *What is the evidence that supports the claimed ???PTEN-independent AKT activation???? Is the observed WDR26-regulated AKT phosphorylation only found in the setting where serum-starved cells are stimulated with the GPCR ligand LPA? LPA works via p110beta which is often activated where PTEN expression is lost. There should be data using insulin or EGF that activate other PI3K isoforms. What is the fate of PTEN activity in the manipulated cells?*

The concept of “PTEN-independent AKT activation” allowed by the lncRNAs was based on the data shown in Supplementary Fig. 6v,w indicating a correlation between high levels of either lncRNA and the levels of phosphorylated AKT (i.e. activated AKT) in a TMA of invasive breast cancer. Since this correlation is also observed in tumours with high expression levels of *PTEN* (and, possibly, have high levels of the *PTEN* protein), we suggested that the AKT phosphorylation induced by the lncRNAs may occur in tumour regardless of *PTEN*. This idea is also supported by the pro-oncogenic activity of *TROLL-2* and *TROLL-3* found both in a melanoma model expressing *PTEN* (Malme-3M cells, ATCC HTB-64) and a melanoma model where *PTEN* is silenced (A375 cells, ATCC CRL-1619) (see Figures 16, also included in the manuscript as new Fig. 5p-w). Since our data is correlative, we have now modified the text and removed any reference to our “PTEN-independent AKT activation” hypothesis.

Regarding the stimulus used to trigger AKT phosphorylation, we used LPA because of a previous report showing that LPA, but not EGF, causes AKT phosphorylation via WDR26 (Ye et al., *Oncotarget*. 2016;7(14):17854-17869).

6. *In addition, the manuscript includes a potential technical error, which should be clarified by the authors before the work is considered further. The authors utilized a doxycycline-inducible CRISPR/Cas9 tool to ???knock down??? genes of interest including those encoding non-coding RNA TROLL-2 (RPSAP52) and TROLL-3 (TRAF3IP2-AS1). They demonstrate that doxycycline induction led to the appearance of approx. 50% indel of the targeted loci (Fig. S2h), reduced RNA signal in derived tumors (Fig. S2i-j), and importantly, an apparent inhibition of tumorigenesis (Fig. 2e), tumor growth (Fig. 2f), and lung metastasis (Fig. 2g-h).*

*Unfortunately it remains difficult to understand how small indel generated by a single guide RNA, which in this case targeting transcription start sites of the two lncRNAs, could produce a functional knockdown or knockout of the targeted non-coding RNA genes (Note that established methods to CRISPR out non-coding RNAs is typically based on a pair of guides to produce large-fragment deletion). In addition, regarding the observed indel (Fig. S2h), it is also unclear how indel could be generated in cells introduced with sgTROLL-3, given that the guide sequence is not followed by the NGG PAM motif for *S. pyogenes* Cas9.*

(The first 23 nucleotides of human TRAF3IP2-AS1 (NR_034108.1) are as follows

cggaaggggc ggcggagcac CGA. Note that the first 20 nts were used as the guide for gene editing by the authors ??? as described in the Method section, and ???CGA??? does not follow the NGG PAM motif used by Cas9).

We would like to thank the reviewer for highlighting this error. Based on this input, we decided to remove from the manuscript all the experiments performed with the CRISPR/Cas9 system to target TROLL-2 and TROLL-3. We instead used shRNAs derived from the siRNAs listed in Supplementary Table 2 (shTROLL-2 is based on siTROLL-2 #1, while shTROLL-3 is based on siTROLL-3 #3), to perform the in vivo experiments assessing the role of TROLL-2 and TROLL-3 in two different xenograft models of breast cancer (CA1D and MDA MB-231 cells) as well as two different models of lung adenocarcinoma (H1299 and

H358 cells) and two of melanoma (A375 and Malme-3M). These new data are shown above as Figures 14-16 and are also included in the manuscript as Fig. 2g-n and Fig. 5h-w. We feel that these additional data further build the case of the oncogenic roles of *TROLL-2* and *TROLL-3*.

REVIEWER COMMENTS

Reviewer #1 (Remarks to the Author):

The authors have addressed most of my suggestions and the manuscript is improved. However, authors have not addressed a couple of important suggestions and these suggestions should be addressed.

Comment 1: Authors provide previous literature to support pro-tumorigenic functions of MALAT1. However, considering recent clinical and experimental analyses of MALAT1 as tumor suppressor in breast cancer and other cancers (Kwok et al., 2018; Kim et al., 2018), it would be good to check whether MALAT1 harbors pro-tumorigenic functions following TAP63 KD.

Comment 4: I do not agree with the authors' explanation. It is a very important aspect of the manuscript to realize the common functions that are controlled by TROLL-2/-3/WDR26 functions. Considering the authors' answer to comment 7 that TROLL-2/-3/WDR26 collectively form a trimeric complex, it would be interesting to know what kind of common biological functions this trimeric complex regulates. One would expect to see common biological functions between TROLL-2, TROLL-3 and WDR26 KDs if three partners are part of the same complex.

ISH: I am not sure where I should look for the signal. Maybe authors should highlight the lncRNA signal using arrows. As I mentioned, single molecule FISH would have been a good option. It seems there is intense nucleolar staining in both control and KD samples. Authors could also support this data using biochemical fractionation.

Reviewer #2 (Remarks to the Author):

Nature Communications manuscript NCOMMS-19-06126A-Z

The response by the group of Elsa Flores is exhaustive. The added data further support a novel axis involving TAP63/lncRNAs and AKT in the control of metastasis of breast cancer as well as other cancers. The correlative (very exhaustive) and partially molecular data shows that TROLL-2 and -3 appear to alter AKT signalling by sequestering the scaffold protein WDR26 in the cytoplasm. The resulting perturbed signalling affects migration and metastasis (although very distinct processes) of the cancer cells.

The work adds to the literature, providing a further mechanism for the readers...it is not clear what is the "real" role of p63 in breast cancer, comparing its effects on TROLLs' lncRNA, Sonic Hedgehog (doi:10.1073/pnas.1500762112; doi:10.1126/scisignal.aaa1033), Frizzled7-Wnt (doi:10.1038/ncb3040), Itch (doi:10.1073/pnas.0603449103; doi:10.1016/j.ccr.2008.06.001) and other miRs (doi:10.1073/pnas.1112257109). But this remains to the reader to discriminate.

No further requests.

Reviewer #3 (Remarks to the Author):

As before there is a great deal of excellent and interesting data in this manuscript. We have two overriding remaining concerns: one is a logical question while the other is technical

Logically we question the title of the paper: The authors conclude "Pan-cancer analysis reveals TAp63-regulated oncogenic lncRNAs (TROLLs) that promote cancer progression through AKT activation". However, there is a lack of any data demonstrating that AKT activation may or may not mediate the phenotypes observed following the alteration of lncRNAs expression. Such experiments might include testing if pan or isoform specific AKT knockdown or inhibition has the same phenotypes as seen from TROLL2/3 knockdown or may block the functional consequences deriving from overexpressing TROLL-2 and TROLL-3 – for example in terms of cell migration and invasion as they showed in Figure 3c, d.

Technically, in response to our previous comments as well as those from Reviewer 1 on the technical validity of CRISPR knockdown of TROLLs, the authors stated that all in vitro and in vivo data based on lncRNA knockdown via CRISPR were in error, and have been replaced with experiments using doxycycline-inducible shRNA to knock down the expression of TROLL-2 and TROLL-3. After carefully reviewing the data as well as consulting with bioinformatics professionals, we find the siRNA/shRNA sequences used to be confusing and in need of clarification by the authors, since they are the foundation for a large series of in vivo assays:

1. The authors listed three sequences for siRNA to target TRAF3IP2-AS1 (TROLL-3), and one of them was chosen for constructing an inducible shRNA vector. However, none of the three siRNA sequences listed could be aligned to any of the 4 RefSeq transcript variants of TROLL-3 (NR_034108.1, NR_034109.1, NR_034110.1, NR_034111.1) (<https://www.ncbi.nlm.nih.gov/gene/643749>) { Note the siRNA sequences for targeting Human TROLL-3(TRAF3IP2-AS1) (from supplemental Table 2) are : GAAACAACUCCACUCCA, GGAAUUGAUAGCCUAUAAA, and GAGCAUUUCCAAUGGAUUA }

We would like to see the following explanations/new data before publication

- 1 The authors need to clarify which RefSeq they used for designing RNAi, and whether the shRNA/siRNA used do actually target the existing isoforms of TRASF3IP2-AS1 in their cells.
2. The identity of doxycycline-inducible shRNA vector, the sequence they used for constructing shControl, and details of shRNA cloning are all lacking and need to be disclosed in Methods section.
3. The oligo probe for in situ hybridization of TROLL-3 (5'-ACTATTACTGCTAACTAATTATGGA-3'), does not completely align to any of the RefSeq transcript variants. The authors need to clarify how they designed the probe.
4. Quantification of lncRNA expression following inducible shRNA expression. The authors should use qPCR to verify that the shRNA does decrease targeted lncRNA expression following an acute induction – for example, a few days of doxycycline treatment in vitro. The only data as in supplemental Figure 2h used in situ hybridization to look at TROLL-2, TROLL-3 expression in xenografts after 10 weeks of doxycycline-supplemented food, providing quite poor confirmation on the efficiency of lncRNA-targeting shRNAs.
5. For overexpression of TROLL-3, the authors constructed a retroviral vector and were able to reach a 100-fold increase of expression as measured by qPCR. The authors need to disclose the sequence of their TROLL-3 cDNA, and clarify whether it corresponds to any of the RefSeq transcript variants.

Response to Reviewers' Comments:

We would like to thank all the reviewers for their thoughtful and thorough review. We have now incorporated all suggestions and feel that the manuscript is greatly improved. Our specific point-by-point response to each of their comments is below in boldface type.

Reviewers' comments:

Reviewer #1 (Remarks to the Author):

The authors have addressed most of my suggestions and the manuscript is improved. However, authors have not addressed a couple of important suggestions and these suggestions should be addressed.

Comment 1: Authors provide previous literature to support pro-tumorigenic functions of MALAT1. However, considering recent clinical and experimental analyses of MALAT1 as tumor suppressor in breast cancer and other cancers (Kwok et al., 2018; Kim et al., 2018), it would be good to check whether MALAT1 harbors pro-tumorigenic functions following TAP63 KD.

As stated in our previous response to the reviewer's comment, the current data obtained by multiple laboratories on the role of human Malat1 in breast cancer are indicative of its function as an oncogene (Huang et al., Oncotarget. 2016 Jun 21;7(25):37957-37965; Wang et al., Breast Cancer Res Treat. 2018 Sep;171(2):261-271; and Zheng et al., Biosci Rep. 2019 Feb 15;39(2). pii: BSR20181284. doi: 10.1042/BSR20181284). Notably, our results showing both higher levels of mouse Malat1 in the absence of the tumour suppressor TAp63 (see Fig. 1a and 1c) and higher levels of human Malat1 in breast cancer cells compared to their normal counterpart (see Fig. 1b and Supplementary Fig. 1a) are in line with the above-mentioned observations. However, we agree with the reviewer that it would be interesting to further investigate Malat1's roles in the absence of TAp63, but – being it beyond the scope of this manuscript – it will be the focus of a future project.

Comment 4: I do not agree with the authors' explanation. It is very important aspect of the manuscript to realize the common functions that are controlled by TROLL-2/-3/WDR26 functions.

Considering the authors answer to comment 7 that TROLL-2/-3/WDR26 collectively form a trimeric complex, it would be interesting to know what kind common biological functions this trimeric complex regulates. One would expect to see common biological functions between TROLL-2, TROLL-3 and WDR26 KDs if three partners are part of the same complex.

Both our *in vitro* and *in vivo* data indicate that one of the common functions of the trimeric complex is clearly the induction of the PI3K/AKT pathway, as judged by AKT phosphorylation on serine 473. We do not exclude that WDR26 and the two lncRNAs may also have additional functions in common, and we aim to further characterize them in the near future.

ISH: I am not sure where I should look for the signal. May be authors should highlight the lncRNA signal using arrows. As I mentioned, single molecule FISH would have been good option. It seems there is intense nucleolar staining exist both control and KD samples. Authors could also support this data using biochemical fractionation.

We appreciate the reviewer's suggestion and we have now added arrows to the ISH panels (Supplementary Fig. 2l,m, Supplementary Fig. 5g',h' and 5m',n') to help the reader focus on the lncRNA signal (purple staining present in the cytoplasm) rather than on the pink signal of the Nuclear Fast Red, which was used as a counterstain.

Reviewer #2 (Remarks to the Author):

Nature Communications manuscript NCOMMS-19-06126A-Z

The response by the group of Elsa Flores is exhaustive. The added data further support a novel axis involving TAp63/lncRNAs and AKT in the control of metastasis of breast cancer as well as other cancers. The correlative (very exhaustive) and partially molecular data shows that TROLL-2 and -3 appear to alter AKT signalling by sequestering the scaffold protein WDR26 in the cytoplasm. The resulting perturbed signalling affects migration and metastasis (although very distinct processes) of the cancer cells.

The work adds to the literature, providing a further mechanism for the readers...it is not clear what is the "real" role of p63 in breast cancer, comparing its effects on TROLLs' lncRNA, Sonic Hedgog (doi:10.1073/pnas.1500762112; doi:10.1126/scisignal.aaa1033), Frizzeld7-Wnt (doi:10.1038/ncb3040), Itch (doi:10.1073/pnas.0603449103; doi:10.1016/j.ccr.2008.06.001) and other miRs (doi:10.1073/pnas.1112257109). But this remains to the reader to discriminate.

No further requests.

We would like to thank the reviewer for his/her comments and for deeming our response to be exhaustive.

Reviewer #3 (Remarks to the Author):

As before there is a great deal of excellent and interesting data in this manuscript. We have two overriding remaining concerns: one is a logical question while the other is technical. Logically we question the title of the paper: The authors conclude “Pan-cancer analysis reveals *TAp63*-regulated oncogenic *lncRNAs* (*TROLLs*) that promote cancer progression through *AKT* activation”. However, there is a lack of any data demonstrating that *AKT* activation may or may not mediate the phenotypes observed following the alteration of *lncRNAs* expression. Such experiments might include testing if pan or isoform specific *AKT* knockdown or inhibition has the same phenotypes as seen from *TROLL2/3* knockdown or may block the functional consequences deriving from overexpressing *TROLL-2* and *TROLL-3* “for example in terms of cell migration and invasion as they showed in Figure 3c, d.

First of all, we would like to thank the reviewer for considering our data interesting and excellent. Following the reviewer’s suggestion, we have now shown that *TROLL-2* and *TROLL-3* function through activation of *AKT*. As suggested, we performed a cell migration and invasion assay to test the effect of *AKT* inhibition in CA1D cells overexpressing either *TROLL-2* or *TROLL-3*. As shown in Figure 1 (also included in the manuscript as new Fig. 6g,h and Supplementary Fig. 6l-n), neither *lncRNA* can promote migration or invasion in cells treated with the *AKT* inhibitor MK-2206, indicating that *AKT* activation is necessary to mediate the oncogenic functions of these two *lncRNAs*.

Figure 1. Cell migration (top left) and invasion (top right) assays in CA1D cells transfected with the indicated

constructs, and treated with either 100 nM MK-2206 or DMSO as control. qRT-PCR for *TROLL-2* and *TROLL-3* (bottom left) and representative western blot of the indicated proteins (bottom right) in the same CA1D cells shown above. Data are mean \pm SD and analysed with two-tailed Student's *t* test, $n = 3$, * vs. pBabe Empty siControl, $P < 0.005$.

Technically, in response to our previous comments as well as those from Reviewer 1 on the technical validity of CRISPR knockdown of TROLLs, the authors stated that all in vitro and in vivo data based on lncRNA knockdown via CRISPR were in error, and have been replaced with experiments using doxycycline-inducible shRNA to knock down the expression of TROLL-2 and TROLL-3. After carefully reviewing the data as well as consulting with bioinformatics professionals, we find the siRNA/shRNA sequences used to be confusing and in need of clarification by the authors, since they are the foundation for a large series of in vivo assays:

1. The authors listed three sequences for siRNA to target TRAF3IP2-AS1 (TROLL-3), and one of them was chosen for constructing an inducible shRNA vector. However, none of the three siRNA sequences listed could be aligned to any of the 4 RefSeq transcript variants of TROLL-3 (NR_034108.1, NR_034109.1, NR_034110.1, NR_034111.1) (<https://www.ncbi.nlm.nih.gov/gene/643749>) { Note the siRNA sequences for targeting Human TROLL-3(TRAF3IP2-AS1) (from supplemental Table 2) are : GAAACAACUCCACUCCA, GGAAUUGAUAGCCUAUAAA, and GAGCAUUUCCAAUGGAUUA}

We would like to see the following explanations/new data before publication

1 The authors need to clarify which RefSeq they used for designing RNAi, and whether the shRNA/siRNA used do actually target the existing isoforms of TRASF3IP2-AS1 in their cells.

When we discovered *TROLL-3* (also known as TRAF3IP2-AS1), we investigated its possible transcripts via the Ensembl Genome Browser, which provides a score (called Transcript Support Level or TSL) that is indicative of the reliability of the information available for each transcript. We first focused our attention on the transcript having the best score (i.e. TSL1), which was ENST00000532226.1, and used it to design 3 siRNAs as well as an ISH probe (see comment #3 below). As it often happens with poorly characterized transcripts,

over the years more information became available and four transcript variants were listed as NCBI Reference Sequences (RefSeq), which are – as indicated by the reviewer – NR_034108.1, NR_034109.1, NR_034110.1, and NR_034111.1. As mentioned at page 10 of our manuscript, NR_034110.1 “is the isoform that we identified as differentially expressed in the MCF10A breast cancer progression model” and we have now corrected the text in the same page clarifying that it is one of the transcripts of *TROLL-3*, and not the longest as we erroneously wrote. NR_034110.1 was used to design novel siRNAs and ISH probe, which were utilized to perform all the experiments reported in our manuscript. We want to thank the reviewer for noticing that we had not listed the updated reagents, and we have now emended the manuscript as follows:

a) siRNAs listed in the Supplementary Table 5:

1. 5'-GAGCAUCAUUUAGAAGAGG-3' (position in NR_034110.1 = 304-322 nt);
2. 5'-GCUGGUCACAAACUCCUG-3' (position in NR_034110.1 = 1617-1635 nt);
3. 5'-GCTATGCAGGATTGGCAGG-3' (position in NR_034110.1 = 88-106 nt).

b) ISH probe listed in the Methods at page 31:

1. 5'-TCGGCGAGGCAAGTGTGAGCA-3' (position in NR_034110.1 = 115-13nt).

2. *The identity of doxycycline-inducible shRNA vector, the sequence they used for constructing shControl, and details of shRNA cloning are all lacking and need to be disclosed in Methods section.*

As requested by the reviewer, the information regarding the doxycycline-inducible shRNA vector, the shRNA negative control (shNT), and the cloning details has been added to the Methods at page 26.

3. *The oligo probe for in situ hybridization of TROLL-3 (5'-ACTATTACTGCTAACTAACTTATGGA-3'), does not completely align to any of the RefSeq transcript variants. The authors need to clarify how they designed the probe.*

As addressed in response to comment #1 (see above), the sequence of the ISH probe has been corrected.

4. Quantification of lncRNA expression following inducible shRNA expression. The authors should use qPCR to verify that the shRNA does decrease targeted lncRNA expression following an acute induction for example, a few days of doxycycline treatment in vitro. The only data as in supplemental Figure 2h used in situ hybridization to look at TROLL-2, TROLL-3 expression in xenografts after 10 weeks of doxycycline-supplemented food, providing quite poor confirmation on the efficiency of lncRNA-targeting shRNAs.

As requested by the reviewer, we have now added to the manuscript the qRT-PCR data regarding the efficiency of the shRNAs in targeting the respective lncRNA after 3 or 6 days of doxycycline treatment *in vitro*. The new data appear as new Supplementary Fig. 2h-k (CA1D and MDA MB-231 cells), new Supplementary Fig. 5c'-f' (H1299 and H358 cells) and 5i'-l' (A375 and Malme-3M cells).

5. For overexpression of TROLL-3, the authors constructed a retroviral vector and were able to reach a 100-fold increase of expression as measured by qPCR. The authors need to disclose the sequence of their TROLL-3 cDNA, and clarify whether it corresponds to any of the RefSeq transcript variants.

As requested by the reviewer, we have now specified in the Methods at page 26 that the NCBI Reference Sequence transcript variant that we overexpressed is NR_034110.1.

REVIEWERS' COMMENTS:

Reviewer #1 (Remarks to the Author):

I have gone through the revised version and authors have not addressed my main criticisms. Authors need to show genes that are commonly regulated on a global scale in order to claim TROLL-2/-3/WDR26 as a functional entity.

I have a major concern with regard to author's presentation of siRNA data on TROLL-3 variants. I looked into TROLL-3 siRNA sequences and the transcript variants used for functional analyses in revision 2 and revision 3 based on reviewer 3 criticisms. In the revision 2, the applicant used siRNAs (GAAACAACUCCACUCCA, GGAAUUGAUAGCCUAUAAA, and GAGCAUUUCCAAUGGAUUA) against TROLL-3 variant ENST00000532226.1. In revision 3, the authors claim that they have used a new set of siRNAs (1. 5'-GAGCAUCAUUUAGAAGAGG-3'; 2. 5'-GCUGGUCACAAACUCCUG-; 3. 5'-GCTATGCAGGATTGGCAGG-3' for another variant (NR_034110.1) of TROLL-3. It is not clear to me how the authors can present functional data for different variants in each revision. These two variants have common 5 exon but 3' exons are different. These issues to need to be look into prior to any decision is made on the manuscript.

Reviewers' comments:

Reviewer #1 (Remarks to the Author):

I have gone through the revised version and authors have not addressed my main criticisms. Authors need to show genes that are commonly regulated on a global scale in order to claim TROLL-2/-3/WDR26 as a functional entity.

Our protein micro-array approach allowed us to identify WDR26 as a protein interacting with both TROLL-2 and TROLL-3, and our *in vitro* and *in vivo* experiments demonstrate that WDR26 and both lncRNAs form a trimeric complex activating the PI3K/AKT pathway. Additionally, our experiments performed using the AKT inhibitor, MK-2206, prove that AKT phosphorylation and subsequent activation are necessary steps in mediating the pro-oncogenic functions of the TROLL-2/TROLL-3/WDR26 trimeric complex.

I have a major concern with regard to author's presentation of siRNA data on TROLL-3 variants. I looked into TROLL-3 siRNA sequences and the transcript variants used for functional analyses in revision 2 and revision 3 based on reviewer 3 criticisms. In the revision 2, the applicant used siRNAs (GAAACAACUUCCACUUCCA, GGAAUUGAUAGCCUAUAAA, and GAGCAUUUCCAAUGGAUUA) against TROLL-3 variant ENST00000532226.1. In revision 3, the authors claim that they have used a new set of siRNAs (1. 5'-GAGCAUCAUUUAGAAGAGG-3'; 2. 5'-GCUGGUCACAAACUUCCUG-;3. 5'-GCTATGCAGGATTGGCAGG-3' for another variant (NR_034110.1) of TROLL-3. It is not clear to me how the authors can present functional data for different variants in each revision. These two variants have common 5 exon but 3' exons are different. These issues to need to be look into prior to any decision is made on the manuscript.

None of the experiments presented in any of the versions of our paper (NCOMMS-19-06126C, NCOMMS-19-06126A-Z, and NCOMMS-19-06126) was performed using the sequences of siRNAs, shRNA, and ISH probe for TROLL-3 erroneously listed in the methods of NCOMMS-19-06126 and NCOMMS-19-06126A-Z. The corrected sequences of

the reagents utilized in all of our experiments and in all versions of the manuscript are indicated in the methods of the current version – NCOMMS-19-06126C. These sequences were used in all experiments of all versions of the manuscript (NCOMMS-19-06126C, NCOMMS-19-06126A-Z, and NCOMMS-19-06126).